# Generic synthesis of small-sized hollow mesoporous organosilica nanoparticles for oxygen-independent X-ray-activated synergistic therapy

Wenpei Fan[1], Nan Lu[2,3], Zheyu Shen[1], Wei Tang[1], Bo Shen[4], Zhaowen Cui[5], Lingling Shan[1], Zhen Yang [1], Zhantong Wang[1], Orit Jacobson[1], Zijian Zhou[1], Yijing Liu[1], Ping Hu[5], Weijing Yang[1], Jibin Song[1], Yang Zhang[6], Liwen Zhang[6], Niveen M. Khashab [6], Maria A. Aronova[7], Guangming Lu[3] & Xiaoyuan Chen [1]

The success of radiotherapy relies on tumor-specific delivery of radiosensitizers to attenuate hypoxia resistance. Here we report an ammonia-assisted hot water etching strategy for the generic synthesis of a library of small-sized (sub-50 nm) hollow mesoporous organosilica nanoparticles (HMONs) with mono, double, triple, and even quadruple framework hybridization of diverse organic moieties by changing only the introduced bissilylated organosilica precursors. The biodegradable thioether-hybridized HMONs are chosen for efficient co-delivery of $tert$-butyl hydroperoxide (TBHP) and iron pentacarbonyl (Fe(CO)$_5$). Distinct from conventional RT, radiodynamic therapy (RDT) is developed by taking advantage of X-ray-activated peroxy bond cleavage within TBHP to generate •OH, which can further attack Fe(CO)$_5$ to release CO molecules for gas therapy. Detailed in vitro and in vivo studies reveal the X-ray-activated cascaded release of •OH and CO molecules from TBHP/Fe(CO)$_5$ co-loaded PEGylated HMONs without reliance on oxygen, which brings about remarkable destructive effects against both normoxic and hypoxic cancers.

[1] Laboratory of Molecular Imaging and Nanomedicine, National Institute of Biomedical Imaging and Bioengineering, National Institutes of Health, Bethesda, MD 20892, USA. [2] Department of Radiology, the Second Affiliated Hospital, Zhejiang University School of Medicine, 310000 Hangzhou, Zhejiang, China. [3] Department of Medical Imaging, Jinling Hospital, Medical School of Nanjing University, 210002 Nanjing, Jiangsu, China. [4] Institute of Radiation Medicine, Fudan University, 200032 Shanghai, China. [5] State Key Laboratory of High Performance Ceramics and Superfine Microstructure, Shanghai Institute of Ceramics, Chinese Academy of Sciences, 200050 Shanghai, China. [6] Smart Hybrid Materials Laboratory (SHMs), Advanced Membranes and Porous Materials Center, King Abdullah University of Science and Technology, Thuwal 23955, Saudi Arabia. [7] Laboratory of Cellular Imaging and Macromolecular Biophysics, National Institute of Biomedical Imaging and Bioengineering, National Institutes of Health, Bethesda, Maryland 20892, USA. These authors contributed equally: Wenpei Fan, Nan Lu. Correspondence and requests for materials should be addressed to Z.S. (email: shenzheyu@nimte.ac.cn) or to G.L. (email: cjr.luguangming@vip.163.com) or to X.C. (email: shawn.chen@nih.gov)

As is well-known, chemotherapy has successfully saved and substantially prolonged the lives of millions of cancer patients during the past decades. But undeniably, conventional chemotherapy usually causes serious systemic toxicity with rather low efficacy owing to the nonspecific distribution and rapid clearance of anticancer drugs[1]. Thus, the emerging unconventional chemotherapy on the basis of nanomedicine has realized a paradigm shift in tumor-specific drug delivery via nanoparticles[2,3], which may overcome several inherent issues associated with free drugs[4–6]. Breakthrough in nanomaterial chemistry has given birth to a large library of nanoscale drug delivery systems (DDSs)[7]. Admittedly, an excellent DDS should harbor the collective merits of both organic and inorganic nanoplatforms[8,9], such as high stability, high biocompatibility, and tumor microenvironment (TME)-driven degradation. Additionally, the DDS had better possess a hollow cavity to allow for a large loading capacity of drugs[10–12]. Featuring the hybridization of organic/inorganic silsesquioxane framework through incorporation of diverse organic groups[13–15], hollow mesoporous organosilica nanoparticles (HMONs) may satisfy the above multifaceted demands by breaking the limitations of traditional inorganic mesoporous silica nanoparticles (MSNs)[16]. Retrospectively, the large-sized HMONs over hundreds of nanometers suffer from short blood circulation and poor tumor accumulation[14,17], which underscores the need for small-sized HMONs below 50 nm for considerable tumor accumulation by achieving the win–win between the decreased RES (reticuloendothelial system) uptake and the increased EPR (enhanced permeability and retention) effect[18–20]. Although some methods have been reported for the preparation of HMONs[14,21–23], it is still essential to work out a generic strategy to synthesize a library of sub-50 nm HMONs with multiple framework hybridization of diverse organic moieties, which may broaden their wide applications.

In parallel to chemotherapy, radiotherapy (RT) is another widely used treatment protocol in the clinic, which can precisely force high-energy X-ray radiation to destroy deep-seated orthotopic tumors without depth limitation[24,25]. However, the inherent tumor hypoxia still remains a long-standing challenge for the further improvement of RT[26]. A variety of in situ tumor oxygenation strategies have thus been proposed to overcome this conundrum[27–31], most of which rely heavily on the TME. Given that the effectiveness of RT hinges upon the generation of reactive oxygen species (ROS) through radiolysis[32], the combination of RT and photodynamic therapy (PDT) to increase the ROS yield has been studied. One representative example is the integration of nanoscintillator and semiconductor for X-ray-activated synchronous RT/PDT without oxygen dependence[33]. Besides, current efforts have been made to explore radiosensitizers that can be activated by X-ray to generate ROS directly, the process of which is void of photo-conversion and oxygen involvement. It has been reported that high-energy X-ray radiation can break low-energy chemical bonds to promote drug release[34–36], such as diselenide bond (Se–Se, 172 kJ mol$^{-1}$)[37], we reasonably speculate that the unstable peroxy bond (O–O, 146 kJ mol$^{-1}$) would also be cleavable in the presence of X-ray radiation. Herein, we identify a paradigm of radiosensitizer, *tert*-butyl hydroperoxide (TBHP), in which X-ray preferentially breaks the lower-energy O-O bond rather than the higher-energy C–H (414 kJ mol$^{-1}$)/O–H (464 kJ mol$^{-1}$)/C–O (326 kJ mol$^{-1}$) bonds. The X-ray-activated O–O bond cleavage generates highly toxic hydroxyl radical (•OH) for enhanced RT, which gives rise to a treatment paradigm of radiodynamic therapy (RDT), more effective than RT. Interestingly, this particular RDT process is immune from tissue oxygen dependency. It has been reported that the strong oxidative activity enables •OH to attack and cleave the Mn–CO coordination bond in $Mn_2(CO)_{10}$ (a CO-releasing molecule (CORM)) for releasing CO molecules[38], which leads to CO gas therapy for causing mitochondria exhaustion and cell apoptosis[39,40]. Therefore, the X-ray-activated •OH generation from TBHP for evoking the CO release from CORM bridges the gap between RDT and gas therapy to achieve synergistic therapy, which again poses strict requirements on outstanding DDSs for co-delivery of TBHP and CORM.

In this study, we propose an ammonia-assisted hot water etching strategy for the generic synthesis of a library of small-sized (sub-50 nm) HMONs with multiple organosilica framework hybridization of diverse organic moieties by changing only the bissilylated organosilica precursors (Fig. 1a). For instance, ten types of sub-50 nm HMONs with mono, double, triple, and even quadruple framework hybridization of thioether/phenylene/ethane/ethylene moieties are successfully produced by selective introduction of bis[3-(triethoxysilyl)propyl]tetrasulfide (BTES)/1,4-bis(triethoxysilyl)benzene (BTEB)/bis-(triethoxysilyl)ethane (BTEE)/bis-(triethoxysilyl)ethylene (BTEEE) precursors. As a paradigm, the thioether-hybridized HMONs are chosen for efficient co-delivery of TBHP and CORM owing to their glutathione (GSH)-responsive biodegradability. The cascaded generation of toxic •OH and CO molecules from the well-designed TBHP/CORM co-loaded biodegradable PEGylated HMONs upon X-ray irradiation (Fig. 1b) shows the obvious advantage in destroying both normoxic and hypoxic tumor cells, which demonstrates the potentialities of nanotechnology in realizing the renaissance of conventional oxygen-dependent RT into oxygen-independent X-ray-activated synergistic radiodynamic/gas therapy.

## Results and discussion

**Generic synthesis of small-sized HMONs**. Traditionally, large-sized HMONs were prepared through a structural difference-based acid/alkaline etching strategy (Fig. 1a)[41]. When depositing a thin mesoporous organosilica (MON) shell on a large-sized Stöber-based dense silica (Supplementary Fig. 1a, b), the Si–C–Si framework within the outer MON shell exhibited higher degree of condensation and stronger chemical stability than the pure Si–O–Si framework within the inner SiO$_2$ core, so the SiO$_2$ core was selectively etched away by HF or Na$_2$CO$_3$, leaving a huge hollow cavity (Supplementary Fig. 1c, d)[42]. Such HMONs were too large (over 150 nm, Supplementary Fig. 2) to be well-dispersed in water (Supplementary Fig. 3). After surface modification with PEG, the circulation half-life of large-sized HMON-PEG in vivo was only 10.8 min (Supplementary Fig. 4a), which was about five-fold less than that of small-sized HMON-PEG (48.9 min, Supplementary Fig. 4b). To obtain small-sized (sub-50 nm) HMONs, cetanecyltrimethylammonium chloride (CTAC) was used as a structural-directing/pre-forming agent to fabricate core/shell structured MSN@MON through the co-hydrolysis of BTES and TEOS at 80 °C. During the process, the amount of triethanolamine (TEA) was the key parameter to control the particle size below 50 nm[43]. Based on the chemical homology principle, the MON layer was directly deposited on the MSN core without pre-centrifugation steps (Fig. 1a). It should be noted that the –Si–C– bond within MON exhibited much higher stability than the –Si–O– bond within MSN under hydrothermal conditions, thus the MON shell was more resistant to hot water etching than the MSN core. Based on this, an ammonia-assisted hot water etching strategy was developed to fabricate sub-50 nm HMONs by selectively etching away the MSN core while leaving the intact MON shell. The internal cavity of the sub-50 nm HMONs enabled substantial encapsulation of various kinds of drugs and other biomolecules.

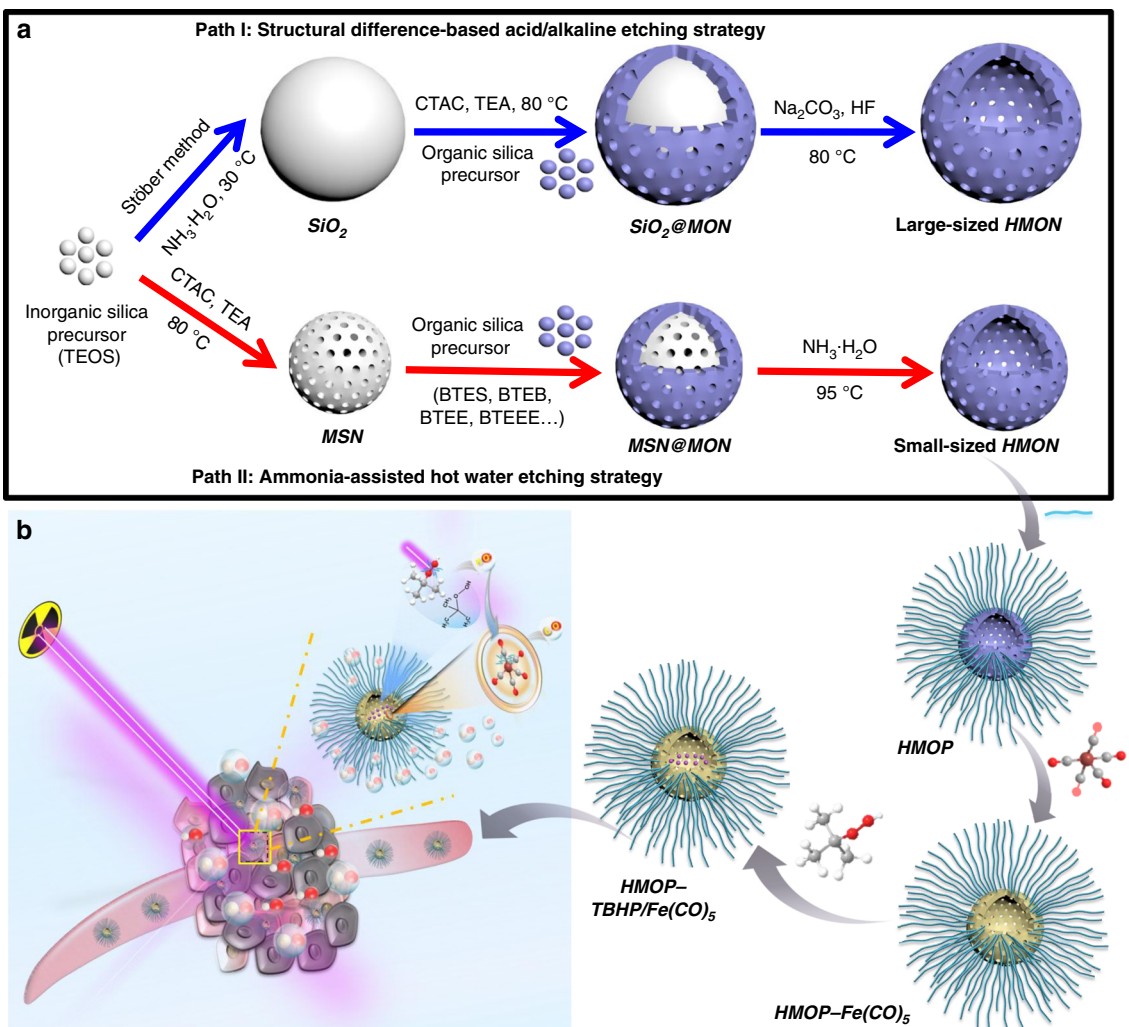

**Fig. 1** Synthesis and biomedical application of HMOP-TBHP/Fe(CO)₅. **a** Schematic of the different synthetic paths for large-sized HMON and small-sized HMON through a structural difference-based acid/alkaline etching strategy and an ammonia-assisted hot water etching strategy, respectively. Herein, a versatile method is first proposed for the generic synthesis of a library of sub-50 nm HMONs with multiple framework hybridization of diverse organic moieties by changing only the introduced bissilylated organosilica precursors. **b** Schematic of the construction of HMOP-TBHP/Fe(CO)₅ for X-ray-activated synergistic radiodynamic/gas therapy. Through PEGylation and co-delivery of TBHP/Fe(CO)₅ by taking advantage of the hollow mesoporous structure of HMON, the cascaded generation of •OH and CO molecules based on the X-ray-activated sequential bond cleavage will give rise to synergistic radiodynamic/gas therapy with extremely little oxygen reliance, thus overcoming the Achilles' heel of conventional radiotherapy

To verify the advantage of this ammonia-assisted hot water etching strategy in the generic synthesis of a library of sub-50 nm HMONs with multiple framework hybridization, four kinds of representative bissilylated organosilica precursors (BTES with thioether moiety, BTEB with phenylene moiety, BTEE with ethane moiety, BTEEE with ethylene moiety) were adopted to form the outer MON layer. As an example, BTES was employed to coat a thioether-hybridized MON shell on the MSN core, and the whole size of the core/shell-structured MSN@MON was kept below 50 nm by increasing the amount of TEA to 0.1 g (Supplementary Fig. 5). The hollow-structured HMONs with an average diameter of 41.8 nm (Fig. 2a, k) were obtained after the MSN core was etched away in hot ammonia solution for 3 h at 95 °C. The emerging Raman shifts at 438/488 cm⁻¹ were ascribed to the specific stretching vibrations of the disulfide bond (Fig. 3k), which indicated the successful framework hybridization of thioether moiety (Fig. 3a). Alternatively, HMONs with phenylene (or ethane or ethylene) hybridization were fabricated by the deposition of a phenylene (or ethane or ethylene)-incorporated MON layer and sequential etching of the MSN core

(Supplementary Figs. 6–8). The only difference was that the outer MON layer was formed through the hydrolysis of BTEB (or BTEE or BTEEE) instead of BTES, and the diameters of all the yielded phenylene (or ethane or ethylene)-hybridized HMONs (Figs. 2b–d, 3b–d, l–n) were less than 50 nm (Fig. 2l–n). According to the similar procedure, sub-50 nm thioether/phenylene (or phenylene/ethane or ethane/ethylene) double-hybridized HMONs (Figs. 2e–g, o–q, 3e–g, o–q, and Supplementary Figs. 9–11) were produced through the co-hydrolysis of equal amounts of BTES/BTEB (or BTEB/BTEE or BTEE/BTEEE) under the catalysis of 0.1 g TEA. Moreover, by co-administering three and four kinds of bissilylated precursors (BTES/BTEB/BTEE, BTES/BTEB/BTEEE, BTES/BTEB/BTEE/BTEEE) while keeping other synthetic parameters unchanged, triple and quadruple-hybridized HMONs (Figs. 2h–j, r–t, 3h–j, r–t, and Supplementary Figs. 12–14) with framework-incorporated multiple moieties (thioether/phenylene/ethane, thioether/phenylene/ethylene, thioether/phenylene/ethane/ethylene) were successfully yielded. Therefore, it could be concluded that the well-established ammonia-assisted hot water etching strategy can be extended to

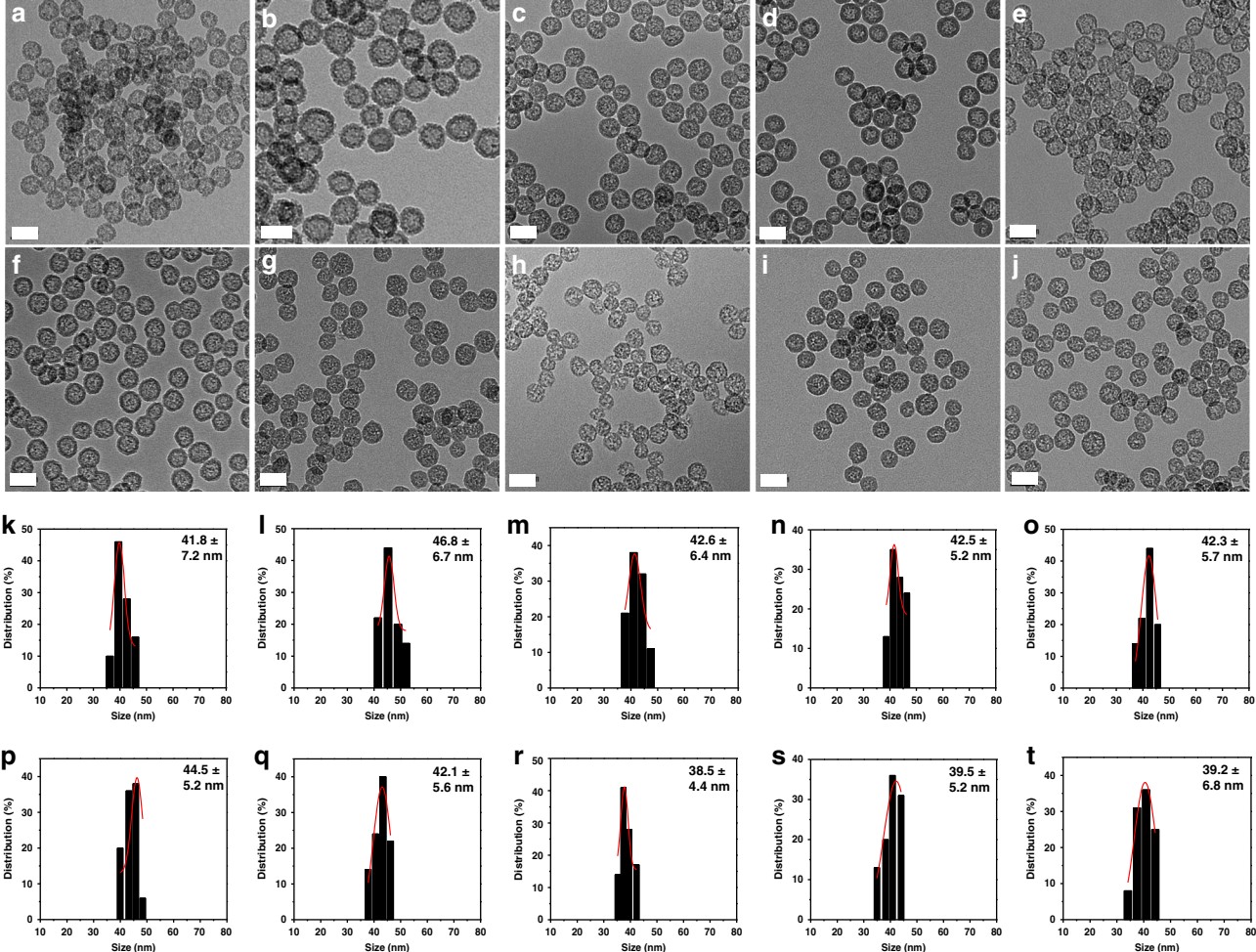

**Fig. 2** Morphology and size distributions of multiple small-sized HMONs. TEM images (**a–j**) and size distributions (**k–t**) of sub-50 nm HMONs with multiple framework hybridization of diverse organic moieties. Mono hybridization: thioether (**a**, **k**), phenylene (**b**, **l**), ethane (**c**, **m**), ethylene (**d**, **n**). Double hybridization: thioether/phenylene (**e**, **o**), phenylene/ethane (**f**, **p**), ethane/ethylene (**g**, **q**). Triple hybridization: thioether/phenylene/ethane (**h**, **r**), thioether/phenylene/ethylene (**i**, **s**). Quadruple hybridization: thioether/phenylene/ethane/ethylene (**j**, **t**). Scale bar: 50 nm. Despite different framework hybridization, the as-synthesized HMONs demonstrate a uniform hollow-structured spherical morphology with size below 50 nm, which confirms the advantage of this well-established ammonia-assisted hot water etching strategy in the generic synthesis of a library of small-sized HMONs

synthesize a library of small-sized multiple-hybridized HMONs with framework incorporation of diverse organic moieties by adjusting only the kind and quantity of bissilylated precursors.

**Characterization of thioether-hybridized HMONs.** The sub-50 nm thioether-hybridized HMON was chosen for the following biomedical applications owing to their reducibility-responsive biodegradability via the GSH-triggered break-up of disulfide bonds within the framework[44]. Despite the temporary appearance of solid-like nanoparticles owing to the dissolved products that might enter and fill the cavity of some non-degraded HMONs or re-generate some silica nanoparticles[45,46], the whole HMONs could be gradually degraded with the increasing time of incubation in GSH solution (Fig. 4a–e, and Supplementary Fig. 15). Especially, the hollow-structured thioether-hybridized HMON exhibited a large surface area of 426 m$^2$ g$^{-1}$ and uniform mesopore sizes of 3–6 nm (Supplementary Fig. 5f, g), allowing for sufficient encapsulation of diverse hydrophilic/hydrophobic payloads.

**Measurement of •OH generation from HMOP-TBHP.** A previously unreported radiosensitizer, TBHP, was loaded into the cavity of HMON through hydrogen bonding force[47]. The low-energy peroxy bond of TBHP was unstable and liable to be damaged by high-energy X-ray radiation, as shown by the methylene blue (MB) bleaching result. The addition of TBHP caused a faster decay of MB absorption under the condition of X-ray radiation (Supplementary Fig. 16), which indicated that •OH generation arose from the cleavage of the O–OH bond of TBHP because MB was usually bleached after selectively trapping •OH[48]. Meanwhile, there is little difference between the •OH yield arising from TBHP + X-ray in normoxic water and that in hypoxic deoxygenated water (Supplementary Fig. 16), which further suggests the oxygen-independent feature of X-ray-activated •OH generation from TBHP.

When 0.65 wt.% TBHP was loaded into PEGylated HMON (HMOP, Supplementary Fig. 17) via vacuum impregnation, we employed terephthalic acid (TA) to measure the •OH yield arising from HMOP-TBHP upon X-ray irradiation. 1.0 mol TA can chemically bind with 1.0 mol •OH radical to produce 1.0 mol 2-hydroxyterephthalic acid (TAOH), so the generated concentration of •OH is equal to that of TAOH[49,50], which can be quantified by measuring its fluorescence emission intensity around 430 nm based on the standard curve (Fig. 4f, g). As HMOP-TBHP itself did not generate •OH and only yielded •OH

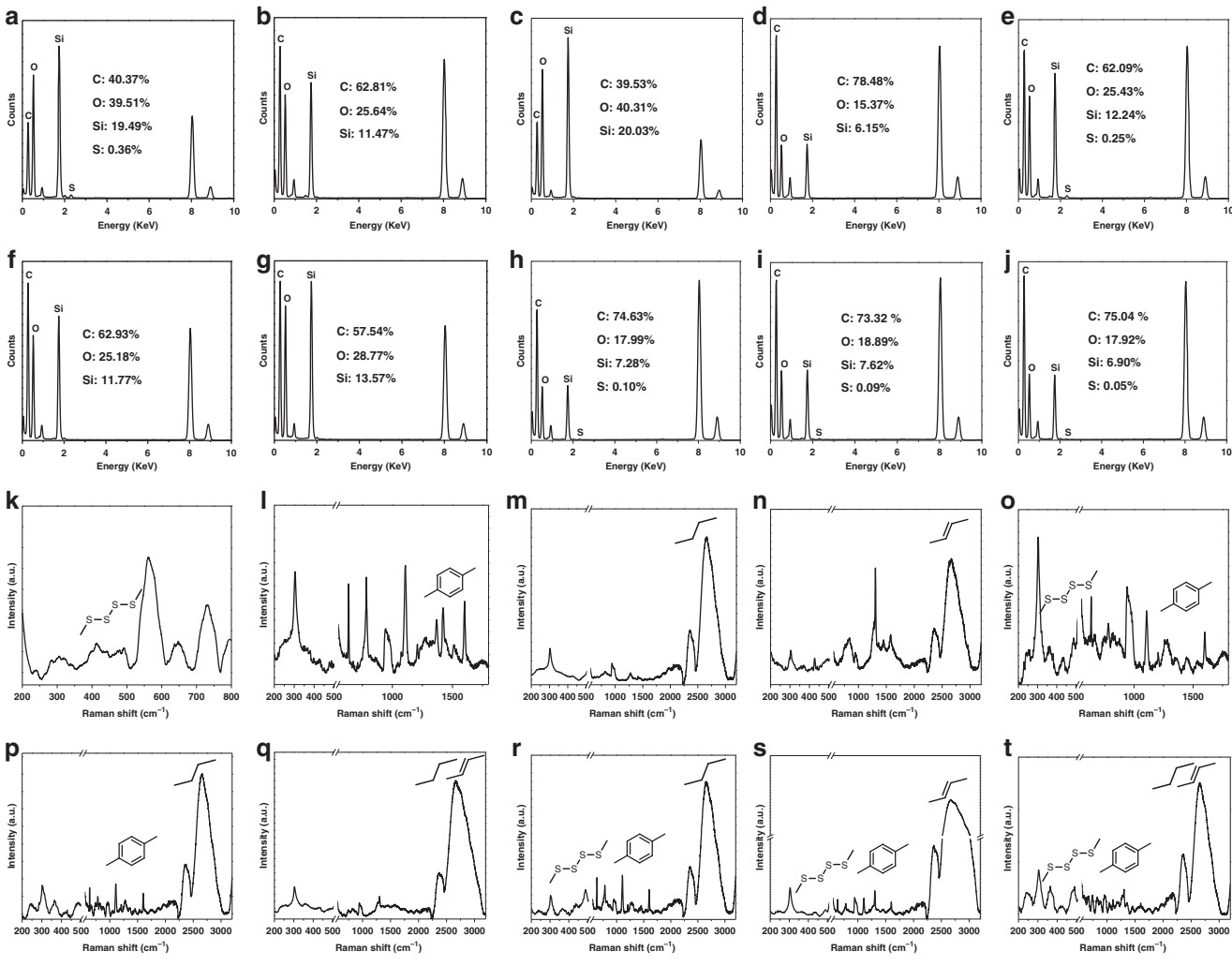

**Fig. 3** Elemental analysis and Raman spectra of multiple small-sized HMONs. Energy-dispersive X-ray spectroscope (EDS) spectra (**a–j**) and Raman spectra (**k–t**) of sub-50 nm HMONs with framework hybridization of different organic moieties. Mono hybridization: thioether (**a**, **k**), phenylene (**b**, **l**), ethane (**c**, **m**), ethylene (**d**, **n**). Double hybridization: thioether/phenylene (**e**, **o**), phenylene/ethane (**f**, **p**), ethane/ethylene (**g**, **q**). Triple hybridization: thioether/phenylene/ethane (**h**, **r**), thioether/phenylene/ethylene (**i**, **s**). Quadruple hybridization: thioether/phenylene/ethane/ethylene (**j**, **t**). Inset in **a–j**: The atomic percentages of the major elements are presented in each EDS spectrum of small-sized HMON. Inset in **k–t**: The representative moieties are presented in each Raman spectrum of small-sized HMON

when X-ray activated the cleavage of the O-OH bond within TBHP, the fluorescent intensity of TAOH was much lower for HMOP-TBHPs without X-ray irradiation than that with X-ray irradiation (Fig. 4h, i). As shown in Fig. 4j, the generated •OH concentration upon 15 Gy of X-ray radiation was calculated to be only 2 nmol mL$^{-1}$ owing to radiolysis of water. However, the addition of HMOP-TBHP (72.2 μmol mL$^{-1}$) could increase the •OH yield to 7 nmol mL$^{-1}$, which suggested the production of 5 nmol mL$^{-1}$ •OH from X-ray-activated HMOP-TBHP. The elevated •OH yield was directly attributed to X-ray-activated breakdown of the O–OH bond in TBHP without any additives, which leads to the naissance of an enhanced RT paradigm, RDT, causing more serious oxidative damage to cancer cells.

**In vitro evaluation of RDT**. To observe the cancer cell uptake of biocompatible HMOP (Supplementary Fig. 18), fluorescein iso-thiocyanate (FITC) was conjugated onto HMOP for both flow cytometry analysis and confocal fluorescence imaging. More uptake of HMOP into the cytoplasm of U87MG cells was observed after prolonged incubation time (Fig. 5a), which was

shown by the increasingly strong green fluorescence signal (from FITC-labeled HMOP) appearing around the blue cell nucleus (Supplementary Fig. 19). After confirming the negligible cyto-toxicity of HMOP-TBHP against U87MG, HepG2, and RAW macrophage cells (Supplementary Figs. 20, 21), we evaluated the intracellular ROS generation by using a fluorogenic probe, 2′,7′-dichlorofluorescein diacetate (DCFH-DA), which could be spe-cifically oxidized by ROS to yield fluorescent DCF[51]. As shown by the quantitative result of flow cytometry (Fig. 5b), although the generated ROS amount in normoxic U87MG cells was increased with the elevated doses of X-ray irradiation owing to the oxygen-involved radiolysis, HMOP-TBHP + RT could significantly raise the intracellular ROS level to a much higher degree. Besides, a much stronger DCF fluorescence signal appeared in normoxic U87MG cells treated by HMOP-TBHP + RT than those subjected to RT alone (Fig. 5c), which further indicated that much higher intracellular ROS yield came from RDT than RT.

The above additional ROS in RDT should cause considerable oxidative damage to DNA, which was evaluated by the comet assay. The damaged DNA exhibited long tail of fluorescent stain, and the degree of DNA damage was determined by the length of

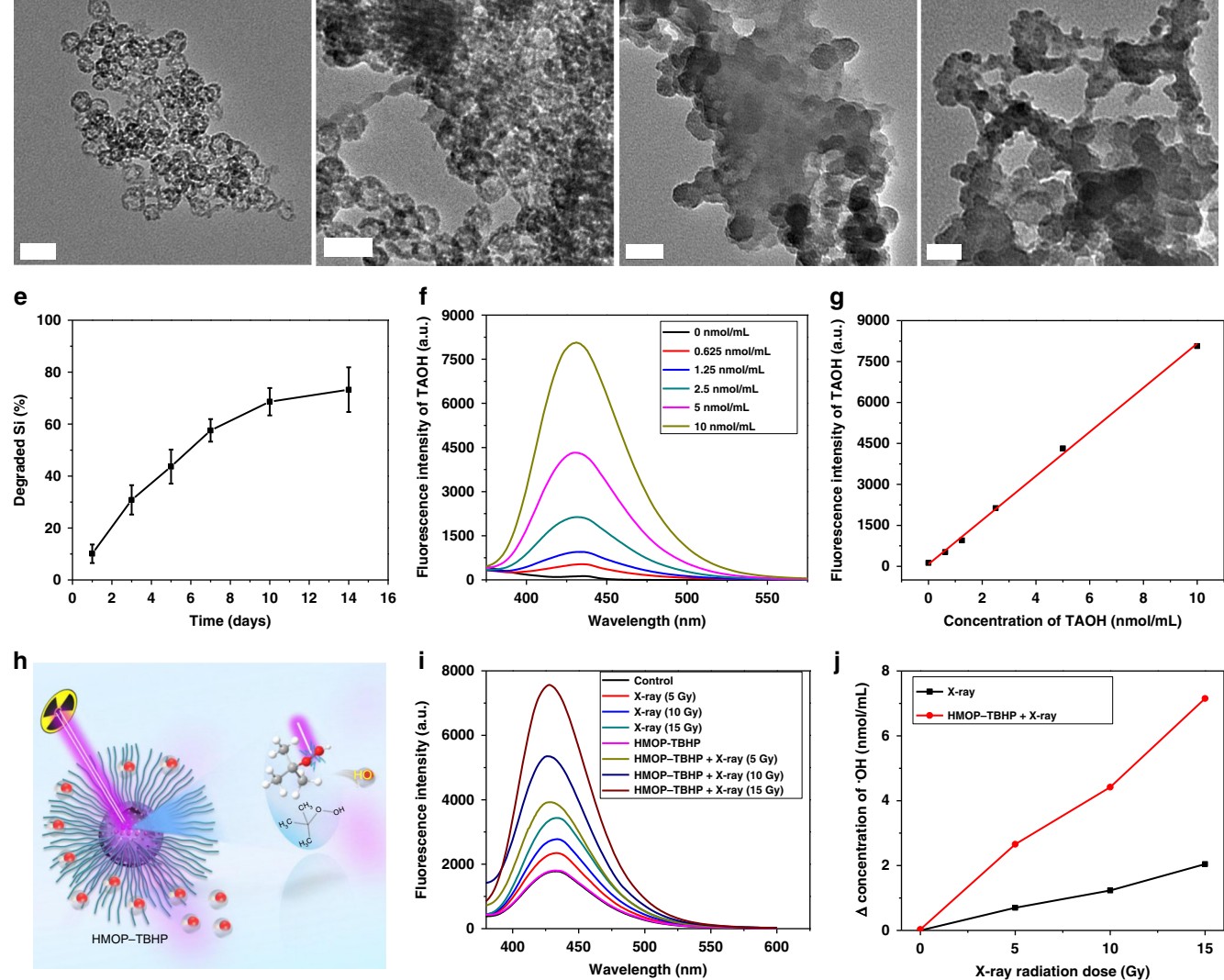

**Fig. 4** Biodegradation of HMON and X-ray-activated •OH generation. **a–d** TEM images of thioether-hybridized HMON dispersed in PBS with 10 mM GSH for 1 day (**a**), 3 days (**b**), 7 days (**c**), and 14 days (**d**). Scale bar: 50 nm. **e** Degradation curve of thioether-hybridized HMON in PBS with 10 mM GSH within 14 days. $n = 3$, mean ± s.d. **f** Fluorescence emission spectra of TAOH with different concentrations. **g** Standard curve for the linear relation between the concentration of TAOH and its fluorescence intensity at $\lambda = 430$ nm. **h** Schematic of X-ray-activated cleavage of peroxy bond within TBHP (encapsulated in the cavity of HMOP) for generation of •OH. **i** Fluorescence spectra of TAOH (oxidized TA) subjected to varied doses (0, 2, 4, 6 Gy) of X-ray irradiation with or without HMOP-TBHP. **j** The •OH concentrations calculated by •OH-triggered oxidation of TA into 2-hydroxyterephalic acid (TAOH)

tail stain. As clearly shown in Fig. 5d, there was no obvious difference in tail DNA stain between the control and RT groups. However, much longer tail DNA stain (with a much higher ratio of tail DNA/head DNA, Fig. 5e) appeared in the HMOP-TBHP + RT group, suggesting the more significant DNA damage caused by RDT than RT. Aggravated DNA damage usually means increased cell death rate and decreased cell viability, which was confirmed by the MTT assay. In contrast to RT, HMOP-TBHP + RT demonstrated much stronger RDT-mediated cell killing effect, which also showed a positive correlation with the X-ray dose and TBHP concentration (Fig. 5f). Supplementary Fig. 22 showed that HMOP had negligible influence on the RT efficacy, thus excluding the possibility of HMOP-mediated radiation enhancement. As the anticancer mechanism of RDT relied on •OH-mediated cleavage of DNA strands to suppress cell proliferation, the survival fractions of normoxic U87MG cells were measured by the colony formation assay at day 15 post treatment. The addition of HMOP-TBHP remarkably intensified the radiation

enhancement effect on reducing the cell survival to an extremely low degree in response to the elevated X-ray doses (Fig. 5h), which unambiguously verified the superior anticancer efficacy of RDT over RT. The excess •OH in RDT could induce irreversible oxidative damage to DNA/protein/lipid for permanently inhibiting cancer cell reproduction.

It should be emphasized that RDT results from X-ray-activated TBHP for additional •OH production without oxygen participation, which represents a noteworthy stride over conventional oxygen-dependent RT. Therefore, the oxygen-independent RDT should take effect under hypoxic condition as well. The ROS level in hypoxic (1% $O_2$) U87MG cells incubated with HMOP-TBHP was also measured through flow cytometry analysis (Supplementary Fig. 23) and confocal fluorescence imaging (Supplementary Fig. 24). As expected, HMOP-TBHP + RT considerably increased the ROS yield in hypoxic U87MG cells although the intracellular ROS generation was suppressed upon RT alone. Accordingly, both RT and HMOP + RT (Supplementary Fig. 25) had negligible

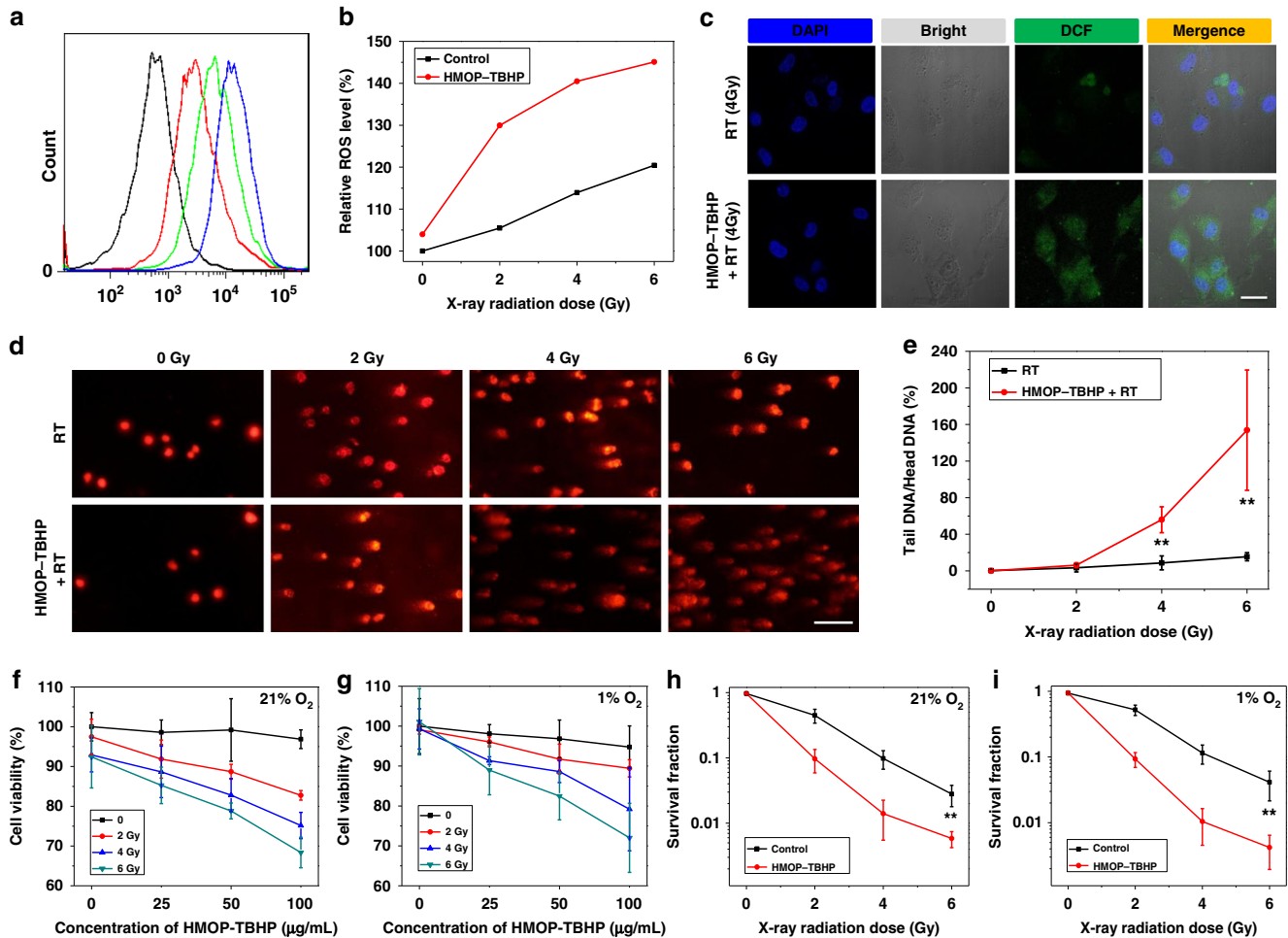

**Fig. 5** In vitro evaluation of RDT. **a** Flow cytometry analysis of U87MG cells after incubation with FITC-labeled HMOP for 1 h (red line), 3 h (green line), and 6 h (blue line). The control U87MG cells received no treatment (black line). **b** Quantitative evaluation of ROS generation in U87MG cells subjected to varied doses (0, 2, 4, and 6 Gy) of X-ray irradiation in the presence or absence of HMOP-TBHP. The intracellular ROS was monitored using a fluorogenic DCFH-DA probe. **c** Confocal fluorescence imaging of normoxic (21% $O_2$) U87MG cells subjected to 4 Gy of X-ray irradiation in the presence or absence of HMOP-TBHP. Scale bar: 50 μm. **d** Fluorescent DNA-stained images (by comet assay) of U87MG cells after treated with or without HMOP-TBHP upon varied doses of X-ray irradiation. Scale bar: 200 μm. **e** Evaluation of DNA damage (ratio of tail DNA/head DNA) of U87MG cells subjected to different treatments. $n = 3$, mean ± s.d., **$P < 0.01$, Student's two-tailed $t$-test. **f, g** Cell viabilities (by MTT assay) of normoxic (21% $O_2$) U87MG cells (**f**) and hypoxic (1% $O_2$) U87MG cells (**g**) subjected to varied doses (0, 2, 4, and 6 Gy) of X-ray irradiation in the presence or absence of different concentrations of (0, 25, 50, and 100 μg mL$^{-1}$) HMOP-TBHP. $n = 4$, mean ± s.d. **h, i** Survival fraction (by colony formation assay) of normoxic (21% $O_2$) U87MG cells (**h**) and hypoxic (1% $O_2$) U87MG cells (**i**) after treated with or without HMOP-TBHP upon varied doses of X-ray irradiation. $n = 3$, mean ± s.d., **$P < 0.01$, Student's two-tailed $t$-test

influence on the hypoxic cell viability at day 1 post treatment while the presence of HMOP-TBHP remarkably raised the hypoxic cell death rate upon elevated doses of X-ray irradiation (Fig. 5g). Moreover, HMOP-TBHP + RT produced an outstanding longtime inhibitory effect on hypoxic cell proliferation (Fig. 5i), which further confirmed the advantage of RDT in combating cancer without reliance on oxygen. Taken together, the oxygen-independent •OH generation on the basis of X-ray-activated peroxy bond cleavage strategy made the well-established RDT technology effective under both normoxic and hypoxic conditions, thus overcoming the Achilles' heel of RT.

**Design and characterization of HMOP-TBHP/Fe(CO)₅.** Single treatment usually fails to completely kill cancer cells owing to its intrinsic limitation, which has driven researchers to develop the technology of multimodal synergistic therapy to maximize the therapeutic effectiveness based on the cooperative enhancement interactions between several treatments[18]. By harnessing the

advantage of highly oxidative •OH radicals in triggering the cleavage of metal-CO bond within CORM to release CO molecules[38], the gap between RDT and gas therapy may be bridged to give rise to X-ray-activated synergistic RDT/gas therapy.

Herein, a typical CORM, iron pentacarbonyl (Fe(CO)₅), was firmly loaded into the mesopores and cavity of HMON through hydrophobic-hydrophobic interactions (Supplementary Fig. 26)[38,42,52]. After PEGylation, the resulting HMOP-Fe(CO)₅ retained uniform spherical morphology (Fig. 6a) and showed high stability with little leakage (Supplementary Figs. 27, 28). The Fe signal that appeared in the elemental mapping (Fig. 6b) and EDS spectrum (Fig. 6c), along with two peaks of Fe orbitals centered at 711 eV ($2P_{3/2}$) and 724 eV ($2P_{1/2}$) in the XPS spectrum (Fig. 6d, e) confirmed the successful attachment of Fe(CO)₅ to HMOP. According to the TG analysis, the loading capacity of Fe(CO)₅ in HMOP was about 3.2 wt% (Fig. 6f) while that in MSN@MON was only 1.5 wt% (Supplementary Fig. 29), which confirmed that hollow silica nanoparticles could load more hydrophobic drug

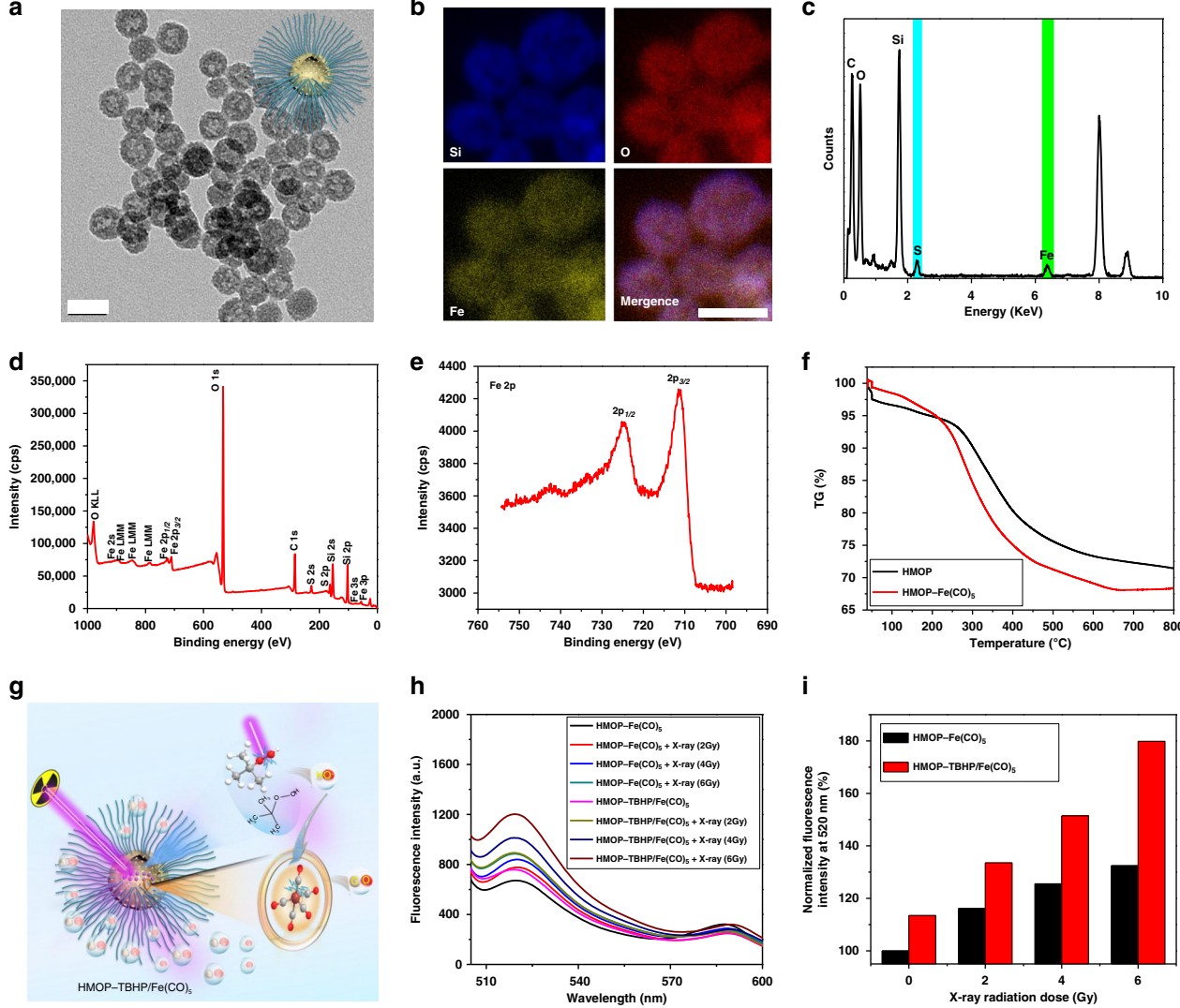

**Fig. 6** Construction of HMOP-TBHP/Fe(CO)$_5$ and X-ray-activated CO release. **a** TEM image of HMOP-Fe(CO)$_5$. Scale bar: 50 nm. **b** Elemental mapping (Si, O, Fe) of HMOP-Fe(CO)$_5$. Scale bar: 50 nm. **c** EDS spectrum of HMOP-Fe(CO)$_5$. **d** X-ray photoelectron spectroscopy (XPS) spectrum of HMOP-Fe (CO)$_5$. **e** Binding energies of Fe($2P_{3/2}$) and Fe($2P_{1/2}$) correspond to 711 and 724 eV, respectively, indicating the attachment of Fe(CO)$_5$ to HMOP. **f** Thermo-gravimetric (TG) curves of HMOP and HMOP-Fe(CO)$_5$. **g** Schematic of X-ray-triggered CO release from HMOP-TBHP/Fe(CO)$_5$. **h** Fluorescence spectra of COP-1 probe subjected to varied doses (0, 2, 4, and 6 Gy) of X-ray irradiation in the presence of HMON-Fe(CO)$_5$ or HMOP-TBHP/Fe(CO)$_5$. **i** Evaluation of CO generation based on the normalized fluorescence intensity of COP-1 probe at 520 nm

molecules than non-hollow ones[53,54]. Despite the narrowed pore size (around 3.1 nm) owing to the partial occupation of Fe(CO)$_5$, HMOP-Fe(CO)$_5$ still had a large surface area of 420.9 m$^2$ g$^{-1}$ (Supplementary Fig. 30). Moreover, the GSH-responsive biodegradability of HMOP was not affected by the Fe(CO)$_5$ payload (Supplementary Fig. 31), which allowed for the in vivo excretion of HMOP-Fe(CO)$_5$. After 0.65 wt% TBHP was encapsulated into the cavity of HMOP-Fe(CO)$_5$, the CO release from HMOP-Fe(CO)$_5$ and HMOP-TBHP/Fe(CO)$_5$ were measured using a fluorogenic CO Probe 1 (COP-1), whose fluorescence emission intensity (peak at 520 nm) was linearly proportional to the CO concentration. HMOP-TBHP/Fe(CO)$_5$ was found to generate much more CO than HMOP-Fe(CO)$_5$ (Fig. 6h, i), suggesting the contribution of TBHP to the increased CO yield. Interestingly, the generated CO quantity also increased with elevated X-ray dose, presumably due to the fact that higher dose of X-ray radiation activated TBHP to generate a larger amount of •OH which further attacked Fe(CO)$_5$ to release much more CO (Fig. 6g).

**In vitro evaluation of synergistic RDT/gas therapy.** After confirming the low cytotoxicity of HMOP-Fe(CO)$_5$ and HMOP-TBHP/Fe(CO)$_5$ (Supplementary Figs. 32, 33), the intracellular CO release was evaluated under both normoxic (21% O$_2$) and hypoxic (1% O$_2$) conditions. In sharp contrast to the suppressed CO generation in hypoxic U87MG cells treated by HMOP-Fe(CO)$_5$ + RT, the CO release from HMOP-TBHP/Fe(CO)$_5$ gradually increased in response to elevated doses of X-ray irradiation (Supplementary Figs. 34, 35), which might be attributed to the X-ray-activated •OH release from TBHP with little oxygen reliance. Consistent with the intracellular ROS release result, the presence of TBHP also triggered HMOP-Fe(CO)$_5$ to release more CO molecules in X-ray-irradiated normoxic U87MG cells (Fig. 7a, b) because the additional •OH generation (as a result of X-ray-activated O–OH bond cleavage within TBHP) could attack Fe (CO)$_5$ to accelerate CO release. Hence, HMOP-Fe(CO)$_5$ showed an oxygen-dependent CO release profile upon X-ray irradiation, yielding a lower gas therapeutic effect against hypoxic U87MG cells than normoxic U87MG cells (Supplementary Fig. 36). In

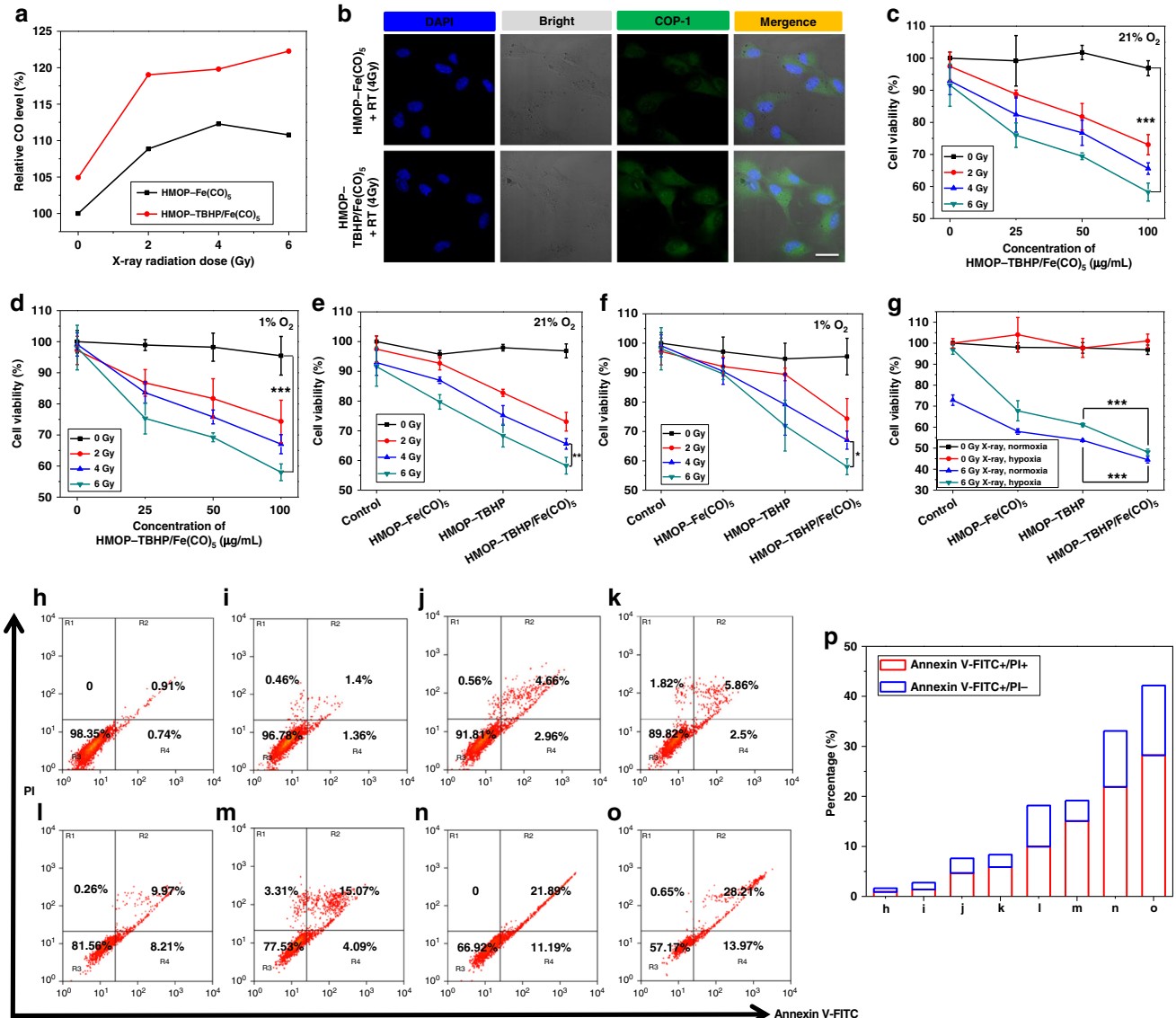

**Fig. 7** In vitro evaluation of synergistic RDT/gas therapy. **a** Quantitative evaluation of CO generation in normoxic (21% $O_2$) U87MG cells after incubation with HMOP-Fe(CO)$_5$ or HMOP-TBHP/Fe(CO)$_5$ upon varied doses (0, 2, 4, and 6 Gy) of X-ray irradiation. The intracellular CO was monitored using a fluorogenic COP-1 probe. **b** Confocal fluorescence imaging of normoxic (21% $O_2$) U87MG cells after incubation with HMOP-Fe(CO)$_5$ or HMOP-TBHP/Fe(CO)$_5$ upon 4 Gy of X-ray irradiation. Scale bar: 50 μm. **c, d** Cell viabilities (by MTT assay) of U87MG cells subjected to varied doses (0, 2, 4, and 6 Gy) of X-ray irradiation with or without different concentrations of (0, 25, 50, 100 μg mL$^{-1}$) HMOP-TBHP/FeCO under normoxic (21% $O_2$) condition (**c**) and hypoxic (1% $O_2$) condition (**d**), respectively. **e, f** Cell viabilities (by MTT assay) of normoxic (21% $O_2$) U87MG cells (**e**) and hypoxic (1% $O_2$) U87MG cells (**f**) after treated with HMOP-Fe(CO)$_5$, HMOP-TBHP or HMOP-TBHP/Fe(CO)$_5$ plus varied doses (0, 2, 4, and 6 Gy) of X-ray irradiation for 24 h. **g** Cell viabilities (by MTT assay) of normoxic and hypoxic U87MG cells after treated with HMOP-Fe(CO)$_5$, HMOP-TBHP, or HMOP-TBHP/Fe(CO)$_5$ plus 6 Gy of X-ray irradiation for 48 h. $n = 4$, mean ± s.d., *$P < 0.05$, **$P < 0.01$, ***$P < 0.001$, Student's two-tailed $t$-test. **h-o** Flow cytometry analysis of the apoptosis of normoxic (21% $O_2$) U87MG cells subjected to different treatments: control (**h**), HMOP-Fe(CO)$_5$ (**i**), HMOP-TBHP (**j**), HMOP-TBHP/Fe(CO)$_5$ (**k**), RT (6 Gy) (**l**), HMOP-Fe(CO)$_5$ + RT (6 Gy) (**m**), HMOP-TBHP + RT (6 Gy) (**n**), HMOP-TBHP/Fe(CO)$_5$ + RT (6 Gy) (**o**). **p** Quantitative analysis of the corresponding cell apoptosis (Annexin V-FITC+/PI−)/necrosis (Annexin V-FITC+/PI+) percentages based on **h-o**

contrast, the X-ray responsive •OH/CO release without oxygen dependence was observed for HMOP-TBHP/Fe(CO)$_5$, which exerted similar synergistic killing effects against both nomoxic and hypoxic cancer cells (Fig. 7c, d).

By comparing the viabilities of U87MG cells treated with TBHP-loaded, Fe(CO)$_5$-loaded, or TBHP/Fe(CO)$_5$ co-loaded HMOP upon varied doses of X-ray irradiation, we observed that HMOP-TBHP/Fe(CO)$_5$ + RT produced much stronger killing effects than HMOP-Fe(CO)$_5$ + RT and HMOP-TBHP + RT, which suggested that the X-ray-activated cascaded release of •OH and CO resulted in a much higher cancer cell death rate

than •OH or CO alone (Fig. 7e–g). Although the effectiveness of HMOP-Fe(CO)$_5$ + RT relied on oxygen concentration, both HMOP-TBHP and HMOP-TBHP/Fe(CO)$_5$ effectively killed normoxic and hypoxic cells upon X-ray irradiation (Fig. 7e, f), which coincided with the above results of intracellular ROS/CO release. To reveal the anticancer mechanism of •OH-mediated RDT, CO-mediated gas therapy, and their synergistic therapy, annexin V-FITC/PI dual-staining assay was performed on normoxic and hypoxic U87MG cells after the corresponding treatments. As seen from Fig. 7h–p and Supplementary Fig. 37, the released CO molecules from X-ray-activated HMOP-Fe(CO)$_5$

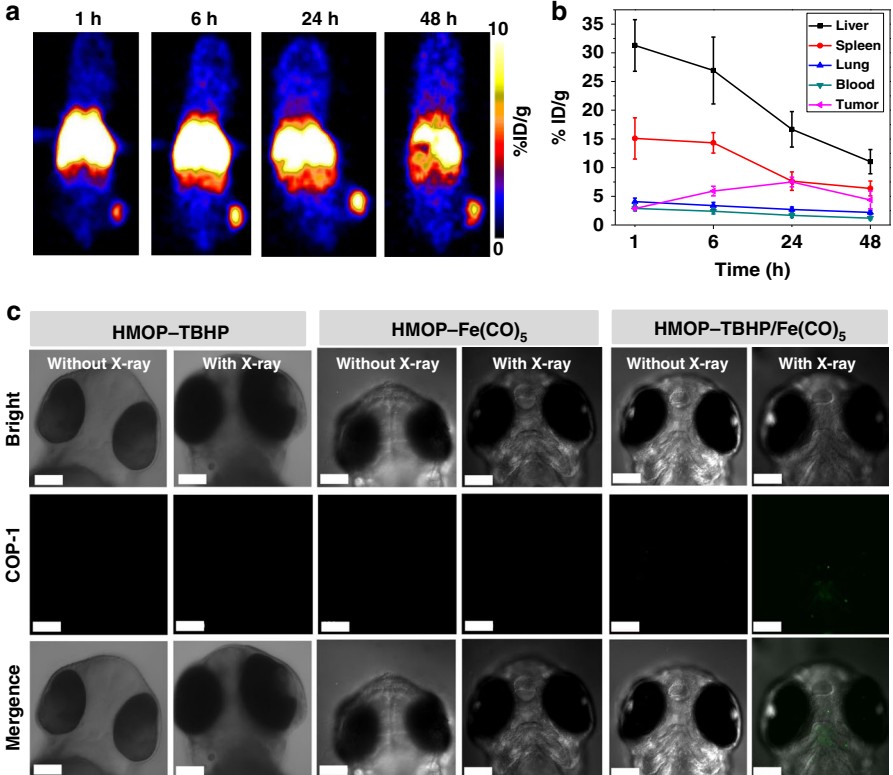

**Fig. 8** Biodistribution of HMOP and in vivo X-ray-activated CO release. **a** PET imaging of U87MG tumor bearing mice at 1, 6, 24, and 48 h after intravenous injection of $^{64}$Cu-labeled HMOP. **b** Biodistribution of HMOP in liver, spleen, lung, blood, and tumor at 1, 6, 24, and 48 h post-injection of $^{64}$Cu-labeled HMOP. $n = 5$, mean ± s.d. **c** Confocal fluorescence imaging of CO generation in living zebrafish larvae with brain microinjection of HMOP-TBHP, HMOP-Fe(CO)$_5$, and HMOP-TBHP/Fe(CO)$_5$ with or without 6 Gy of X-ray irradiation, respectively. CO was monitored using a fluorogenic COP-1 probe. Scale bar: 120 μm

mainly caused cell necrosis while the •OH arising from X-ray-activated HMOP-TBHP resulted in cell apoptosis and necrosis. Moreover, the combination of RDT and gas therapy based on the cascaded generation of •OH and CO through X-ray-activated HMOP-TBHP/Fe(CO)$_5$ led to a much higher percentage of apoptotic and necrotic cells (Fig. 7p), suggesting much improved •OH/CO-mediated synergistic co-killing effects. Consequently, on the basis of the well-designed HMOP-TBHP/Fe(CO)$_5$, RDT and gas therapy had the chance to be linked together to revolutionize conventional RT into more effective X-ray-activated synergistic therapy, which could break oxygen dependence and yield significantly higher cancer-combating efficacy than the corresponding individual treatment paradigm (Supplementary Figs. 38, 39).

**In vivo evaluation of synergistic RDT/gas therapy**. Before the in vivo study, the biodistribution of HMOP was evaluated by PET imaging, which would determine the delivery efficiency of TBHP and Fe(CO)$_5$ to the tumor region. Herein, $^{64}$Cu was used to label HMOP through the thiol group owing to its strong chelating affinity towards radionuclides[55], which was confirmed by the extremely high radiochemical yield of almost 100% (Supplementary Fig. 40). Moreover, the obtained $^{64}$Cu-labeled HMOP showed relatively high radiolabeling stability in both PBS and serum (Supplementary Fig. 41), which allowed for the use for PET imaging. After intravenous injection, HMOP was found to quickly lighten the U87MG tumor within 6 h (Fig. 8a), and the highest tumor uptake of HMOP at 24 h postinjection was about 7% ID/g (Fig. 8b), at which X-ray irradiation was applied. The optical transparency of zebrafish larva allowed for observation of

the in vivo CO release under a confocal fluorescence microscope. Contrary to the invisible fluorescence signal in the larval brain after microinjection of HMOP-TBHP or HMOP-Fe(CO)$_5$ regardless of X-ray irradiation, a well-marked green fluorescence signal of the COP-1 probe only appeared in the presence of HMOP-TBHP/Fe(CO)$_5$ plus X-ray irradiation (Fig. 8c), which confirmed the feasibility of X-ray-activated CO release from HMOP-TBHP/Fe(CO)$_5$ in vivo. Of note, systemic injection of HMOP-TBHP/Fe(CO)$_5$ caused no obvious adverse effects on the major organs of mice (Supplementary Fig. 42), thus ensuring the bio-safety of the following in vivo synergistic treatment.

As shown by the tumor growth curves in Fig. 9a, single RT failed to effectively suppress U87MG tumor growth mainly due to the presence of hypoxia in the core of the tumor that attenuated the efficacy of RT. Effective tumor growth inhibition and even regression (Fig. 9b, and Supplementary Fig. 43) were only achieved by the treatment of HMOP-TBHP/Fe(CO)$_5$ + RT rather than HMOP-Fe(CO)$_5$ + RT or HMOP-TBHP + RT. The remarkable cooperative enhancement interactions between RDT and gas therapy produced considerable •OH/CO co-killing effect that was more effective than RDT or gas therapy alone. The satisfactory anticancer efficacy of HMOP-TBHP/Fe(CO)$_5$ + RT stemmed from the significant up-regulation of caspase 3 and p53 (cancer apoptosis-related proteins, Fig. 9c, d, and Supplementary Fig. 44), causing large-scale tumor cell apoptosis, which was verified by the TUNEL staining results (Fig. 9h–n). In agreement with the in vitro therapeutic effects measured by MTT, HMOP-TBHP/Fe(CO)$_5$ + RT resulted in a higher tumor apoptosis ratio than RT or HMOP-Fe(CO)$_5$ + RT, which might be attributed to the superiority of oxygen-independent RDT in killing hypoxic tumor cells over oxygen-reliant RT or RT-induced CO gas therapy. The

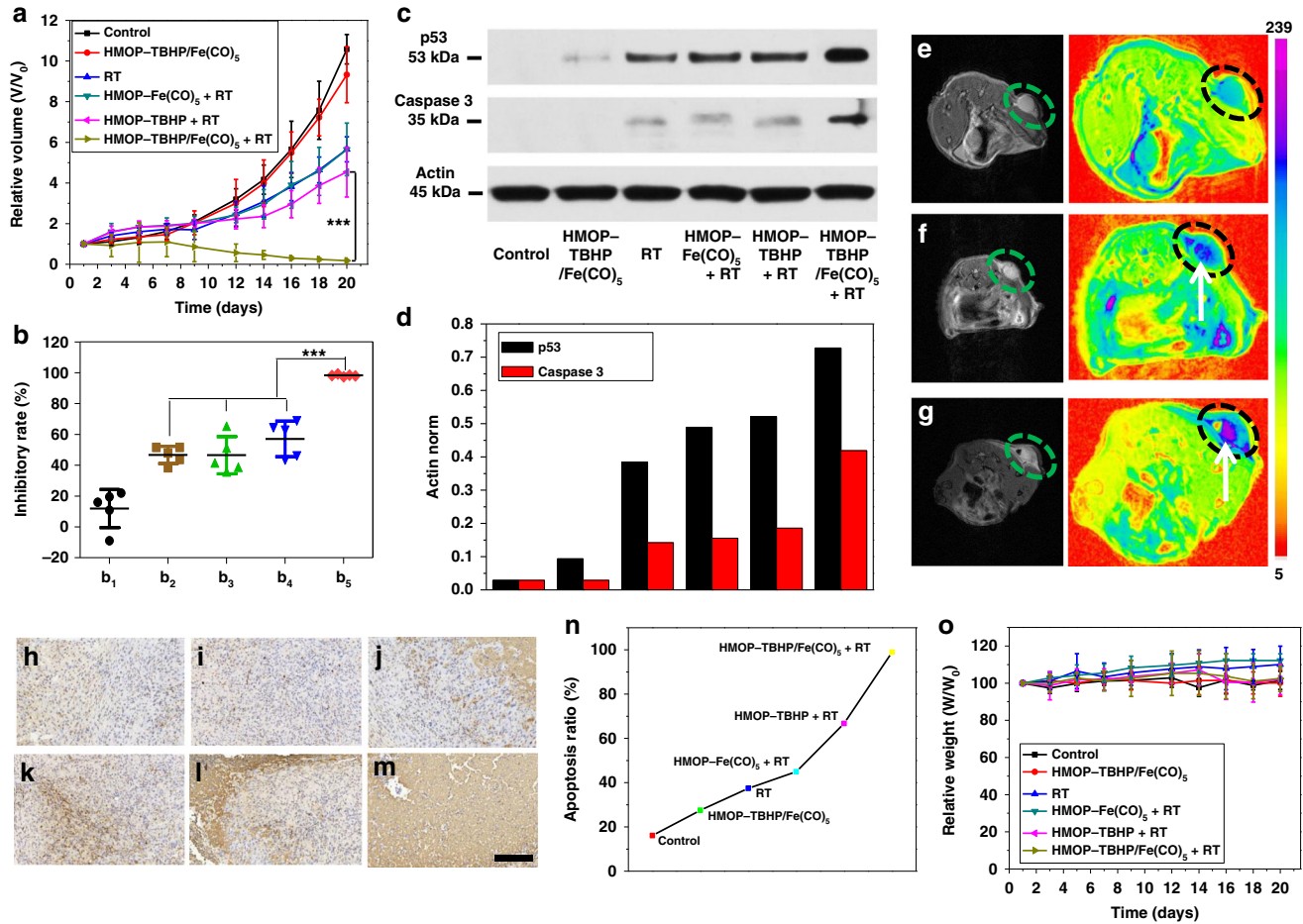

**Fig. 9** In vivo evaluation of synergistic RDT/gas therapy. **a** U87MG tumor growth curves after different treatments. **b** Inhibitory rates of U87MG tumors at day 20 post treatment: HMOP-TBHP/Fe(CO)$_5$ ($b_1$), RT ($b_2$), HMOP-Fe(CO)$_5$ + RT ($b_3$), HMOP-TBHP + RT ($b_4$), HMOP-TBHP/Fe(CO)$_5$ + RT ($b_5$). $n = 5$, mean ± s.d., ***$P < 0.001$, Student's two-tailed $t$-test. **c** Western blot analysis of p53 and caspase 3 expression in U87MG tumors after different treatments. Uncropped image available in Supplementary Fig. 44. **d** The corresponding quantitative analysis of p53 and caspase 3 expression in **c**. **e-g** $T_2$-weighted MR imaging and the corresponding pseudo-color images of U87MG tumors at day 3 post treatment: control (**e**), RT (**f**), HMOP-TBHP/Fe(CO)$_5$ + RT (**g**). The white arrows refer to the apoptotic area in tumor. **h-m** TUNEL-stained sections of U87MG tumors subjected to different treatments: control (**h**), HMOP-TBHP/Fe(CO)$_5$ (**i**), RT (**j**), HMOP-Fe(CO)$_5$ + RT (**k**), HMOP-TBHP + RT (**l**), HMOP-TBHP/Fe(CO)$_5$ + RT (**m**). $n = 5$ sections per group. Scale bar: 200 µm. **n** Quantitative analysis of tumor cell apoptosis in **h-m** by using the Image-pro plus 6.0 software (Media Cybernetics, Inc., MD, USA). **o** Weight change curves of U87MG tumor-bearing mice after different treatments. $n = 5$, mean ± s.d

additional •OH generation in RDT could greatly accelerate the release of CO from Fe(CO)$_5$ with little oxygen dependence that substantially enhanced gas therapy against hypoxic cancer, so the resulted synergistic RDT/gas therapeutic effect achieved over 95% tumor apoptosis by the treatment of HMOP-TBHP/Fe(CO)$_5$ + RT. $T_2$-MR imaging (Fig. 9e–g) and H&E staining (Supplementary Fig. 45) results also showed that HMOP-TBHP/Fe(CO)$_5$ + RT brought forth more remarkable tumor cell apoptosis/necrosis than RT or other individual treatments, which further confirmed the advantage of the well-developed synergistic RDT/gas therapy in destroying malignant hypoxic tumors and breaking the oxygen dependence of most current X-ray-activated treatment protocols. It was anticipated that the feature of X-ray in precise positioning would make normal tissues void of radiation damage, leaving little influence on the health status of mice, which was evidenced by the little fluctuation in the weight of mice after all these X-ray-related treatments (Fig. 9o).

In summary, different from conventional acid/alkaline etching chemistry, an ammonia-assisted hot water etching strategy was developed in this study for the generic synthesis of a library of small-sized (sub-50 nm) HMONs with multiple framework hybridization of diverse organic moieties. In total, ten types of

mono, double, triple, and even quadruple-hybridized HMONs with framework-incorporated diverse moieties (thioether, phenylene, ethane, ethylene) were produced to demonstrate the versatility of such a typical method in synthesizing multiple-hybridized HMONs by changing only the introduced bissilylated organosilica precursors. The biocompatible/biodegradable thioether-hybridized PEGylated HMONs were prepared to first co-deliver TBHP (a radiosensitizer) and Fe(CO)$_5$ (a CORM) via hydrogen bonding force and hydrophobic-hydrophobic interaction, respectively. Intriguingly, the unstable peroxy bond within TBHP was selectively cleaved to generate highly oxidative •OH, which not only directly caused irreversible cancer cell death via an oxygen-dependent RDT process, but also attacked the Fe-CO coordination bond within Fe(CO)$_5$ to accelerate CO release for gas therapy. The cascaded generation of •OH and CO through a previously unexplored X-ray-activated sequential bond cleavage strategy led to synergistic RDT/gas therapy for substantial destruction of both normoxic and hypoxic cancer by breaking the oxygen dependency of conventional RT. This study establishes a versatile synthetic method for sub-50 nm multiple-hybridized HMONs which allow X-ray-activated multimodal synergistic therapy, which is expected to significantly advance the

development of deep penetrating treatment technology without tissue oxygen dependence.

## Methods

**Materials**. Cetyltrimethylammonium chloride (CTAC) solution (25 wt% in $H_2O$), triethanolamine (TEA), tetraethyl orthosilicate (TEOS), bis[3-(triethoxysilyl)propyl]tetrasulfide (BTES), 1,4-bis(triethoxysilyl)benzene (BTEB), bis-(triethoxysilyl)ethane (BTEE), bis-(triethoxysilyl)ethylene (BTEEE), glutathione (GSH), *tert*-butyl hydroperoxide (TBHP) solution (70 wt% in $H_2O$), iron(0) pentacarbonyl (Fe$(CO)_5$), terephthalic acid (TA), 2-hydroxyterephthalic acid (TAOH), 2′,7′-dichlorofluorescein diacetate (DCFH-DA), (3-Mercaptopropyl)trimethoxysilane (MPTES), methylene blue (MB), ammonia solution, dimethyl sulfoxide (DMSO), and NaCl were purchased from Sigma-Aldrich. mPEG2K-silane was purchased from Creative PEGWorks. The CO Probe 1 (COP-1) was purchased from Wuxi AppTec Co., Ltd, China. All reagents were of analytical grade and used without any purification.

**Synthesis of thioether-hybridized HMON**. First, core/shell-structured MSN@MON was constructed based on the chemical homology principle. 2 g of CTAC and 0.1 g of TEA were mixed in 20 mL of water under vigorous stirring for 0.5 h and then transferred to 80 °C oil base. One milliliter of TEOS was added dropwise into the above system for 1 h of reaction, followed by the addition of the mixed silica precursors (1 mL of BTES and 0.5 mL of TEOS) for another 3 h of reaction. Afterward, the resulting MSN@MON particles were obtained by centrifugation and washed with ethanol several times. The CTAC template of MSN@MON was extracted in 30 mL of methanol with NaCl (1 wt.%). The extraction procedure was repeated at least three times to guarantee the complete removal of CTAC. Second, MSN@MON was made into hollow-structured HMON through an ammonia-assisted hot water etching strategy. MSN@MON was dispersed in 30 mL of water and then transferred to 95 °C oil base. After addition of $NH_4OH$, the above system was reacted for 3 h to etch away the inner MSN core. The final thioether-hybridized HMON was obtained by centrifugation and washed with water several times.

**Synthesis of other moieties-hybridized HMON**. The synthetic procedures were similar to the above process for thioether-hybridized HMON except for the introduction of bissilylated organosilica precursors. For phenylene-hybridized HMON, the adopted bissilylated organosilica precursors are the mixture of 1 mL of BTEB and 0.5 mL of TEOS. For ethane-hybridized HMON, the adopted precursors are the mixture of 1 mL of BTEE and 0.5 mL of TEOS. For ethylene-hybridized HMON, the adopted precursors are the mixture of 1 mL of BTEEE and 0.5 mL of TEOS. For thioether/phenylene double-hybridized HMON, the adopted precursors are the mixture of 0.5 mL of BTES and 0.5 mL of BTEB. For phenylene/ethane double-hybridized HMON, the adopted precursors are the mixture of 0.5 mL of BTEB and 0.5 mL of BTEE. For ethane/ethylene double-hybridized HMON, the adopted precursors are the mixture of 0.5 mL of BTEE and 0.5 mL of BTEEE. For thioether/phenylene/ethane triple-hybridized HMON, the adopted precursors are the mixture of 0.5 mL of BTES, 0.5 mL of BTEB, and 0.5 mL of BTEE. For thioether/phenylene/ethylene triple-hybridized HMON, the adopted precursors are the mixture of 0.5 mL of BTES, 0.5 mL of BTEB, and 0.5 mL of BTEEE. For thioether/phenylene/ethane/ethylene quadruple-hybridized HMON, the adopted precursors are the mixture of 0.5 mL of BTES, 0.5 mL of BTEB, 0.5 mL of BTEE, and 0.5 mL of BTEEE.

**Biodegradation evaluation of thioether-hybridized HMON**. To mimic the reductive tumor microenvironment, the biodegradation behavior of HMON was evaluated in both PBS and simulated body fluid (SBF) containing 10 mM GSH. Six milligram of HMON was dispersed in both 30 mL of PBS with 10 mM GSH and 30 mL of SBF with 10 mM GSH for incubation at 37 °C under slow stirring (300 rpm). Both the concentrations of HMON in PBS and SBF were 0.2 mg mL$^{-1}$. After different durations (1, 3, 5, 7, 10, and 14 days) of incubation, 1 mL of PBS and 1 mL of SBF were taken out and centrifuged to collect the partially biodegraded HMON. Subsequently, both the TEM characterization and ICP-AES analysis were performed to observe the degradation behavior of HMON and evaluate the degradation rate, respectively. More details about the evaluation of HMOP-Fe$(CO)_5$ degradation and Fe$(CO)_5$ release can be found in the Supplementary Methods section.

**Synthesis of HMOP (PEG-modified HMON)**. 50 mg of HMON and 30 mg of PEG-silane (Mw = 2000) were mixed in 50 mL of ethanol and then stirred/refluxed at 78 °C overnight. Finally, the yielded HMOP was obtained by centrifugation and washed with ethanol several times. More details about the synthesis of HMOP-TBHP, HMOP-Fe$(CO)_5$, HMOP-TBHP/Fe$(CO)_5$, [64]Cu-labeled HMOP, and the radiolabelling stability test can be found in the Supplementary Methods section.

**Cell lines and cell culture**. U87MG, HepG2, and RAW macrophage cells were purchased from the American Type Culture Collection (ATCC). These cells were cultured in DMEM media with 10% FBS, and 1% penicillin and streptomycin. All the cells were tested to be free of mycoplasma contamination.

**In vitro measurement of intracellular ROS**. The intracellular ROS was measured using a fluorogenic probe for ROS (DCFH-DA). For flow cytometry analysis, normoxic U87MG cells were seeded at several 6-well plates with a density of $10^5$ per well and cultured at 37 °C under 21% $O_2$ for 24 h. Then 10 μM DCFH-DA was added to each plate for 20 min of incubation. After washing away free DCFH-DA, 100 μg mL$^{-1}$ HMOP-TBHP was added to each plate for 6 h of incubation. Then the normoxic U87MG cells were exposed to varied doses (0, 2, 4, and 6 Gy) of X-ray irradiation. Afterward, the cells were harvested, washed and re-suspended in PBS for flow cytometry analysis of the average fluorescent density. For confocal fluorescence imaging, $10^4$ U87MG cells per well were seeded into a 4-well CLSM plate and cultured at 37 °C under 21% $O_2$ for 24 h. 100 μg mL$^{-1}$ HMOP-TBHP was added into the well for 6 h of co-incubation. Then the normoxic U87MG cells were exposed to 4 Gy of X-ray irradiation. Afterward, the cells were washed, stained with DAPI, and fixed for observation on the confocal fluorescence microscope (Zeiss LSM 780).

The procedures of measuring the ROS amount in hypoxic U87MG cells were similar to the above process for ROS measurement in normoxic U87MG cells except for incubating hypoxic U87MG cells at 37 °C under 1% $O_2$.

**In vitro evaluation of RDT**. For MTT, normoxic U87MG cells were seeded into several 96-well plates at a density of $10^4$ cells per well and cultured at 37 °C under 21% $O_2$ for 24 h. Different concentrations (0, 25, 50, and 100 μg mL$^{-1}$) of HMOP-TBHP were added to each well for 6 h of incubation, and then the cells were exposed to varied doses (0, 2, 4, and 6 Gy) of X-ray irradiation. After incubation for another 24 h, the old DMEM media were replaced by 100 μL of DMEM media of MTT (5 mg mL$^{-1}$) for another 4 h of incubation. The MTT in each well was replaced by 100 μL of DMSO, and the absorbance of each well was monitored by a microplate reader at the wavelength of 570 nm. For colony formation assay, U87MG cells were divided into two groups. For the group subject to RT, normoxic U87MG cells were seeded into several 6-well plates at a density of 100, 150, 300, or 400 cells per well and cultured at 37 °C under 21% $O_2$ for 24 h, which were exposed to 0, 2, 4, or 6 Gy of X-ray irradiation, respectively. For the group subject to HMOP-TBHP + RT, normoxic U87MG cells were seeded into several 6-well plates at a density of 500, 1000, 1500, or 2000 cells per well and cultured at 37 °C under 21% $O_2$ for 24 h. After the addition of 100 μg mL$^{-1}$ HMOP-TBHP for 6 h of incubation, the cells were exposed to 0, 2, 4, or 6 Gy of X-ray irradiation, respectively. All the treated cells were incubated for another 15 days, and then the as-formed colonies were fixed and stained with Gimesa. The survival fractions of normoxic U87MG cells in each group were determined by calculating the ratio of colony numbers that contained more than 50 cells.

The procedures of evaluating the RDT effect on hypoxic U87MG cells were similar to the above process for normoxic U87MG cells except for incubation of U87MG cells at 37 °C under 1% $O_2$.

**In vitro measurement of intracellular CO**. The intracellular CO was measured using a fluorogenic CO Probe 1 (COP-1). For flow cytometry analysis, normoxic U87MG cells were seeded at several 6-well plates with a density of $10^5$ per well and cultured at 37 °C under 21% $O_2$ for 24 h. 1 μM COP-1 was added to each plate for 20 min of incubation. After washing away free COP-1, 100 μg mL$^{-1}$ HMOP-Fe$(CO)_5$ or HMOP-TBHP/Fe$(CO)_5$ was added to each plate for 6 h of incubation. Then the normoxic U87MG cells were exposed to varied doses (0, 2, 4, and 6 Gy) of X-ray irradiation. Afterward, the cells were harvested, washed, and re-suspended in PBS for flow cytometry analysis of the average fluorescent density. For confocal fluorescence imaging, $10^4$ U87MG cells per well were seeded into a 4-well CLSM plate and cultured at 37 °C under 21% $O_2$ for 24 h. 100 μg mL$^{-1}$ HMOP-Fe$(CO)_5$ or HMOP-TBHP/Fe$(CO)_5$ was added into the well for 6 h of co-incubation. Then the normoxic U87MG cells were exposed to 4 Gy of X-ray irradiation. Afterwards, the cells were washed, stained with DAPI, and fixed for observation on the confocal fluorescence microscope (Zeiss LSM 780).

The procedures of measuring the CO amount in hypoxic U87MG cells were similar to the above process for CO measurement in normoxic U87MG cells except for incubation of U87MG cells at 37 °C under 1% $O_2$.

**Annexin V-FITC/PI dual-staining assay**. Normoxic U87MG cells were seeded into several 6-well plates at a density of $10^5$ cells per well and cultured at 37 °C under 21% $O_2$ for 24 h. 100 μg mL$^{-1}$ HMOP-Fe$(CO)_5$, HMOP-TBHP, or HMOP-TBHP/Fe$(CO)_5$ was added to each well for 6 h of incubation, and then the cells were exposed to 6 Gy of X-ray irradiation. Then all of the cells were harvested and co-stained with annexin V-FITC and PI for flow cytometry analysis.

The procedures of annexin V-FITC/PI dual-staining assay on hypoxic U87MG cells were similar to the above process for normoxic U87MG cells except for incubation of U87MG cells at 37 °C under 1% $O_2$.

More details about the cell uptake evaluation (by flow cytometry analysis and fluorescence imaging) and in vitro therapy evaluation (by MTT assay, comet assay, and calcein AM/PI dual-staining assay) can be found in the Supplementary Methods section.

**Animal studies.** Healthy female nude mice (4–6 weeks old) were obtained from Harlan Laboratories (Frederick, MD, USA). All animal work was conducted in appliance to the NIH Guide for the Care and Use of Animals under protocols approved by the NIH Clinical Center Animal Care and Use Committee (NIH CC/ ACUC).

**In vivo toxicity evaluation of HMOP-TBHP/Fe(CO)$_5$.** The female nude mice (4–6 weeks old) were intravenously administered with a single dose of HMOP-TBHP/Fe(CO)$_5$ (20 mg mL$^{-1}$, in 150 μL PBS). Several other mice were used as the control. The mice were anesthetized and dissected at 30 days post-injection. The major organs (heart, liver, spleen, lung, and kidney) were dissected, fixed in a 10% formalin solution and stained with hematoxylin and eosin (H & E) for histological analysis.

**In vivo evaluation of synergistic RDT/gas therapy.** U87MG cells ($2 \times 10^6$ cells per site) were implanted subcutaneously into female nude mice (4–6 weeks old). In vivo therapy experiments were carried out when the tumor reached ~6–8 mm in average diameter. The mice were divided into six groups. The first group of mice received PBS, as control group; the second group was intravenously injected with HMOP-TBHP/Fe(CO)$_5$ (20 mg mL$^{-1}$, in 150 μL PBS), as HMOP-TBHP/Fe(CO)$_5$ group; the third group was subjected to 8 Gy of X-ray radiation, as RT group; the fourth group was intravenously injected with HMOP-Fe(CO)$_5$ (20 mg mL$^{-1}$, in 150 μL PBS), and then subjected to 8 Gy of X-ray radiation 24 h later, as HMOP-Fe(CO)$_5$ + RT group; the fifth group was intravenously injected with HMOP-TBHP (20 mg mL$^{-1}$, in 150 μL PBS), and then subjected to 8 Gy of X-ray radiation 24 h later, as HMOP-TBHP + RT group; the sixth group was intravenously injected with HMOP-TBHP/Fe(CO)$_5$ (20 mg mL$^{-1}$, in 150 μL PBS), and then subjected to 8 Gy of X-ray radiation 24 h later, as HMOP-TBHP/Fe(CO)$_5$ + RT group. During three week after the corresponding treatments, the volume of tumors was measured every other day and calculated by the following equation: $V = L \times W^2/2$. The tumors were sectioned into slices for H&E and TUNEL staining analysis. More details about the X-ray irradiation source and characterization methods can be found in the Supplementary Methods section.

**Statistical analysis.** Statistical analysis was performed by using Student's two-tailed *t*-test with statistical significance assigned at *$P < 0.05$ (significant), **$P < 0.01$ (moderately significant), and ***$P < 0.001$ (highly significant).

## Data availability
The main data supporting the findings of this study are available within the article and the Supplementary Information. Extra data are available from the corresponding authors upon reasonable request.

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

## Acknowledgements

We gratefully acknowledge support from the National Natural Science Foundation of China (51602203, 81530054, 51761145021), the Intramural Research Program (IRP) of the NIBIB, NIH, the Youth Innovation Promotion Association of Chinese Academy of Sciences (2016269), and the National Key Research & Development Program (2016YFC1400600, 2018YFD0800300). We also thank Cindy Clark, NIH Library Writing Center, for manuscript editing assistance.

## Author contributions

W.F., Z.S., G.L. and X.C. conceived and designed the project. W.F. and Z.S. synthesized the materials. W.F., Z.C., Z.Y., Z.Z., Y.L., P.H., J.S., Y.Z, L.Z., N.M.K. and M.A.A. performed the material characterizations. N.L. conducted most of the in vitro and in vivo experiments. W.T., L.S. and B.S. helped do the flow cytometry experiment. Z.W. and O.J. performed the PET imaging experiment. W.Y. helped draw the schematic pictures. W.F., Z.S., G.L. and X.C. co-wrote the paper. All of the authors approved the final version.

## Additional information

**Competing interests:** The authors declare no competing interests.

