## [Peer Review File · Nature Communications]

Reviewers' comments:

Reviewer #1 (Remarks to the Author):

The paper "Generic synthesis of small-sized hollow mesoporous organosilica nanoparticles for oxygen-independent X-ray-activated synergistic therapy", by Chen et al, deals with the development of a new treatment paradigm of radiodynamic therapy.

The major claims of the paper are novel and they will be surely of interest for experts in several different fields. The potentialities of nanotechnologies in terms of nanostructuring of the matter and their disruptive power when are shifted in nanomedicine applications are, in the same way, the relevant aspects in this paper. From this point of view, they could also be better emphasized in the abstract, in the last part of the introduction and in the conclusions. The paper can be accepted for the publication in Nature Communications.

Reviewer #2 (Remarks to the Author):

In this work, the authors present a silica nanoparticle together with a new radiosensitizer for oxygen independent radiosensitization.

Major points:

- 1) There seems to not be too much evidence that this radiation sensitizer is oxygen independent.
- 2) It is confusing why two radiosensitizers are used. Actually, in combination with new silica nanoparticle synthesis methods, new radiosensitizers, focus on oxygen-independent radiosensitization, new animal models (zebrafish), there is a lot to digest in this work. I recommend the authors try to simplify their message.
- 3) It seems straightforward to use GC/MS or LC/MS to verify the mechanism of action of the novel photosensitizer
- 4) Some of the language is a cliched. For example, consecutive sentences start with the following:
1) "In addition", 2) "Drawing inspiration", 3) "Of special note", 4) "Exhiliratingly", 5) "Extensively"

Reviewer #3 (Remarks to the Author):

In this study, Fan et. al. employed an "ammonia-assisted hot water etching" strategy for the synthesis of small-sized (sub-50 nm) hollow mesoporous organosilica nanoparticles, and examined their co-delivery performance of tert-butyl hydroperoxide (TBHP) and iron pentacarbonyl (Fe(CO)₅) for oxygen-independent X-ray-activated synergistic therapy. Overall, the experiments were carefully conducted, the data was well analyzed, and the manuscript was well organized. However, there are still some significant problems in the current study that make it not suitable to be published in "Nature Communications".

1> For the "ammonia-assisted hot water etching" strategy highlighted by the authors in this study, actually, this strategy has been developed at least one decade ago for hollow materials synthesis (such as Adv. Mater. 2000, 12, 103–106; Adv. Mater. 2008, 20, 3987–4019; ACS Nano, 2010, 4, 529–539; Adv. Healthcare Mater. 2015, 4, 2797–2801). This is not a new method, the authors did not mention this in the manuscript. And even in the authors' recent paper (ACS Nano, 2018, 12, 1580–1591), they also used a similar method to fabricate sub-50 nm hollow organosilica nanoparticles.

2> For the synthesis of smaller-sized hollow mesoporous organosilica nanoparticles, there have already various one-step methods for hollow mesoporous organosilica nanoparticles synthesis, such as Chem. Commun., 2016, 52, 3544-3547; J. Mater. Sci. 2017, 52, 2868-2878.

Therefore, combined the 1st and 2nd comments, I did not find any new points for the materials synthesis (especially small-sized hollow mesoporous organosilica) in this study.

3> As for the hollow nanoparticles used in this study, the authors mentioned that "hollow cavity to

allow for a large loading capacity of drugs", but actually in this study, the loading capacity of Fe(CO)₅ is only 3.2 wt.%. I am not sure if it is necessary to use hollow nanoparticles in this study. If possible, please provide a comparison of drug loading between hollow and non-hollow nanoparticles.

4> What is the loading capacity of TBHP?

5> Please provide more details of degradation study, such as temperature, particle concentration, TEM sample collection method, etc. And what is the degradation rates as a function of time?

6> Since the particles were administrated by intravenous injection, please also provide the degradation data of HMONS in simulated body fluid or blood fluid.

7> In this study, the authors mentioned the loading details of TBHP and Fe(CO)₅ into HMONS separately, but actually, there is almost no loading information for the resultant HMOP-TBHP/Fe(CO)₅ material, what is the loading amount of TBHP and Fe(CO)₅ in HMOP-TBHP/Fe(CO)₅? What is the detailed loading protocol? And how about the stability of HMOP-TBHP/Fe(CO)₅ since the loading is mainly relying on noncovalent interactions (hydrogen bonding force and hydrophobic-hydrophobic interactions)?

8> Hydroxyl radical ($\cdot\text{OH}$) released from TBHP could further trigger Fe(CO)₅ to release CO molecules, this finding is very interesting in this study. However, as mentioned above, since there is multi-step protocol for producing HMOP-TBHP/Fe(CO)₅, how to keep the amount and ratio of TBHP and Fe(CO)₅ in a controllable, stable, and reproducible way?

9> In figure 7c, the HMOP-TBHP group should be provided.

10> In Figures 3c and S21a, why the primary hollow nanoparticles turned out to be solid ones? Please clarify.

11> In Figure 8c-d, for better comparison, please also provide the results of other treatment groups.

Therefore, based on the above considerations, there is limited new information in the materials synthesis part, the cascaded generation $\cdot\text{OH}$ and CO for multimodal synergistic therapy is a new finding, but the construction of HMOP-TBHP/Fe(CO)₅ particulate system is not clearly presented.

Reviewer #4 (Remarks to the Author):

The authors report synthesis and biological evaluation of small sized hollow mesoporous organosilica nanoparticles (HMONS) encapsulating a radiosensitizer tert-butyl hydroperoxide (TBHP) and iron pentacarbonyl (Fe(CO)₅) for a novel type cancer therapy, which combines radiodynamic therapy (X-ray activated hydroxy radical generation) and a CO gas therapy. Several hydroxy radical-forming radiosensitizers are currently in clinical trials for enhancing efficacy of radiotherapy. In this paper X-ray activated hydroxy radical generation and CO gas therapy are delicately combined into a same nanoformulation in a novel way, leading to excellent tumor growth inhibition when evaluated in U87MG tumor-bearing mice. These results may lead to more efficient anti-cancer therapies and encourage discovery of other dynamic combination treatments. In order to be able to evaluate applicability and significance of the presented results the following comments should be addressed:

-The authors claim reporting a generic method for synthesis of small-sized HMONS, demonstrated by fabrication of three different HMNOs. More comprehensive characterization of the fabricated materials should be presented for facilitating comparison of the different hybridized HMNOs; such as mesopore size for all particle types and elemental analysis for quantification of the organic moieties. The authors are asked to report also the BTES/BTEB ratio on the dual hybridized HMNOs.

-Fe(CO)₅ was adsorbed into the mesoporous channel via weak interactions. In Fig S19 the authors demonstrate stability of the Fe(CO)₅ adsorption by inspecting color of the supernatant. More persuasive evaluation of Fe(CO)₅ leakage should be presented. Please, quantify the possible Fe(CO)₅ release at different time points in the storage buffer and also in serum.

- Figure 3i: The authors should explain why the fluorescence intensity is lower for HMOP-TBHPs even without X-ray irradiation?
- The authors claimed that biosafety of HMOP-TBHP/Fe(CO)₅ was confirmed by evaluating the H&E stained sections of major organs after iv-injection of the particles. The in vivo biodistribution study of ⁶⁴Cu-labeled HMOPs demonstrate that there is major accumulation of the particles in liver and spleen. Cell viability was tested only in U87MG cells (Fig. S22). Cell viability in hepatic cells (e.g. HepG2) and in RAW macrophages should be presented.
- Fig 7b: add %ID/g in lung and blood at different time-points.
- Description of the Cu-64 radiolabeling procedure should be added to the supporting information, including also evaluation of stability of the radiolabel (i.e. possible Cu-64 release) at different time points in PBS (pH 7.4) and serum.
- The authors should report number of animals used for in vivo toxicity evaluation and for each treatment group in evaluation of the therapeutic effect in U87MG tumors. Number of the animals should be included also to all figure captions reporting any in vivo data.
- Line 201 and Fig S8: Explain MB
- Details of the X-ray source and absorbed dose calculation for the cell studies and in vivo experiments should be provided.

Detailed responses to the reviewers' comments

Reviewers' comments:

Reviewer #1 (Remarks to the Author):

The paper "Generic synthesis of small-sized hollow mesoporous organosilica nanoparticles for oxygen-independent X-ray-activated synergistic therapy", by Chen et al, deals with the development of a new treatment paradigm of radiodynamic therapy. The major claims of the paper are novel and they will be surely of interest for experts in several different fields. The potentialities of nanotechnologies in terms of nanostructuring of the matter and their disruptive power when are shifted in nanomedicine applications are, in the same way, the relevant aspects in this paper. From this point of view, they could also be better emphasized in the abstract, in the last part of the introduction and in the conclusions. The paper can be accepted for the publication in Nature Communications.

Response: Thanks for the reviewer's positive comments and strong recommendation for publishing this paper in *Nature Communications*. We have made significant improvements and emphasized the potentialities of nanotechnology in the abstract, the last part of the introduction, and the conclusion.

Reviewer #2 (Remarks to the Author):

In this work, the authors present a silica nanoparticle together with a new radiosensitizer for oxygen independent radiosensitization.

Major points:

1) There seems to not be too much evidence that this radiation sensitizer is oxygen independent.

Response: Thanks for the reviewer's comment. The introduced radiosensitizer, *tert*-butyl hydroperoxide (TBHP), can generate hydroxyl free radical ($\bullet\text{OH}$) without the participation of oxygen for oxygen independent radiosensitization, which has been corroborated by both persuasive theoretical analysis and adequate experiment data in this study. First, a number of literatures (Ref. 31-34) have reported that high-energy X-ray radiation can directly break low-energy chemical bonds to promote drug release, such as diselenide bond (Se-Se, 172 kJ/mol). Within the structure of TBHP, the binding energy of unstable O-O bond (146 kJ/mol) is much lower than that of C-H bond (414 kJ/mol), O-H bond (464 kJ/mol), and C-O bond (326 kJ/mol), so the lower-energy O-O bond can be preferentially cleaved by X-ray to generate hydroxyl radical ($\bullet\text{OH}$). As the whole process of X-ray-activated O-O bond cleavage does not involve the participation of oxygen, the $\bullet\text{OH}$ generation from TBHP upon X-ray irradiation does not rely on oxygen. Second, the methylene blue (MB) dye was used to specifically probe the $\bullet\text{OH}$ generation from X-ray-activated TBHP because it could be bleached after selectively trapping $\bullet\text{OH}$ (Ref. 48). As shown in Fig. S16, there is virtually no difference between the $\bullet\text{OH}$ yield arising from TBHP + X-ray in normoxic water and that in hypoxic deoxygenated water, which indicates that oxygen is not a necessity for the X-ray-activated $\bullet\text{OH}$ generation from TBHP and the process is oxygen independent. Third, the intracellular $\bullet\text{OH}$ (ROS) generation was measured by using a fluorogenic probe, 2',7'-dichlorofluorescein diacetate (DCFH-DA), which could be specifically oxidized by ROS to yield fluorescent DCF (Ref. 51). As shown in Figs. 5b, 5c, S23, and S24,

although the intracellular ROS generation was suppressed upon X-ray irradiation (RT) alone in hypoxic (1% O₂) U87MG cells, the combination of HMOP-TBHP and RT (denoted as HMOP-TBHP + RT) could generate appreciable amount of ROS in both normoxic (21% O₂) and hypoxic (1% O₂) U87MG cells. Meanwhile, the survival fractions of U87MG cells after being treated by HMOP-TBHP + RT under normoxic (21% O₂) and hypoxic (1% O₂) conditions were reduced to almost the same level (Figs. 5f-i), which suggests that the process of intracellular •OH generation from HMOP-TBHP + RT for killing cancer cells is also oxygen independent. Overall, the above theoretical and experimental evidences are sufficient to confirm that TBHP acts as an oxygen-independent radiosensitizer for enhanced X-ray-activated ROS generation and radiosensitization with little reliance on oxygen involvement.

Related references in the revised manuscript:

31. Tanabe, K., Asada, T., Ito, T. & Nishimoto, S.-i. Radiolytic Reduction Characteristics of Drug-Encapsulating DNA Aggregates Possessing Disulfide Bond. *Bioconjugate Chem.* **23**, 1909-1914 (2012).
32. Cao, W., Gu, Y., Meineck, M. & Xu, H. The Combination of Chemotherapy and Radiotherapy towards More Efficient Drug Delivery. *Chem. Asian J.* **9**, 48-57 (2014).
33. Starkewolf, Z.B., Miyachi, L., Wong, J. & Guo, T. X-ray triggered release of doxorubicin from nanoparticle drug carriers for cancer therapy. *Chem. Commun.* **49**, 2545-2547 (2013).
34. Ma, N., *et al.* Radiation-Sensitive Diselenide Block Co-Polymer Micellar Aggregates: Toward the Combination of Radiotherapy and Chemotherapy. *Langmuir* **27**, 5874-5878 (2011).
48. Satoh, A.Y., Trosko, J.E. & Masten, S.J. Methylene Blue Dye Test for Rapid Qualitative Detection of Hydroxyl Radicals Formed in a Fenton's Reaction Aqueous Solution. *Environ. Sci. Technol.* **41**, 2881-2887 (2007).
51. Idris, N.M., *et al.* *In Vivo* Photodynamic Therapy Using Upconversion Nanoparticles As Remote-Controlled Nanotransducers. *Nat. Med.* **18**, 1580-1585 (2012).

2) It is confusing why two radiosensitizers are used. Actually, in combination with new silica nanoparticle synthesis methods, new radiosensitizers, focus on oxygen-independent radiosensitization, new animal models (zebrafish), there is a lot to digest in this work. I recommend the authors try to simplify their message.

Response: Thanks for the reviewer's comment. We think the reviewer might have misunderstood the main idea of this paper. Only one radiosensitizer, *tert*-butyl hydroperoxide (TBHP), is introduced in this study because it can generate highly toxic hydroxyl radical (•OH) upon X-ray irradiation to increase the ROS yield for enhanced RT. However, iron pentacarbonyl (Fe(CO)₅) is not a radiosensitizer but a typical CO-releasing molecule (CORM) because it can only generate CO rather than ROS. The reason for the co-use of TBHP and Fe(CO)₅ is to maximally unlock the innate power of X-ray to sequentially generate •OH and CO for substantially improved therapeutic output. By harnessing the advantage of highly oxidative •OH radicals in triggering the cleavage of Fe-CO bond within Fe(CO)₅ to release CO molecules, the well-developed X-ray-activated synergistic RDT/gas therapy gave rise to much stronger •OH/CO-mediated synergistic co-killing effects than single RDT or CO gas therapy through the use of TBHP or Fe(CO)₅ alone, as clearly shown in Figs. 7, S37-39.

Besides, the context of this study is straightforward to follow. First, a unique versatile "ammonia-assisted hot water etching" strategy is introduced for the generic synthesis of a library of small-sized (sub-50 nm) HMONs with multiple framework hybridization of diverse organic moieties by changing only the bisilylated organosilica precursors. To demonstrate the generality of this synthetic method, ten types of sub-50 nm HMONs with uniform hollow-structured spherical morphology were successfully produced: four types of mono-hybridized HMON with framework incorporation of one kind of moiety (thioether, phenylene, ethane, or ethylene)

through the use of BTES, BTEB, BTEE or BTEEE precursor; three types of double-hybridized HMON with framework incorporation of two kinds of moieties (thioether/phenylene, phenylene/ethane, or ethane/ethylene) through the use of mixed BTES/BTEB, BTEB/BTEE, or BTEE/BTEEE precursors; two types of triple-hybridized HMON with framework incorporation of three kinds of moieties (thioether/phenylene/ethane or thioether/phenylene/ethylene) through the use of mixed BTES/BTEB/BTEE or BTES/BTEB/BTEEE precursors; quadruple-hybridized HMON with framework incorporation of four kinds of moieties (thioether/phenylene/ethane/ethylene) through the use of mixed BTES/BTEB/BTEE/BTEEE precursors. Second, a typical paradigm of sub-50 nm biodegradable thioether-hybridized HMON was chosen to co-deliver TBHP and $\text{Fe}(\text{CO})_5$ for X-ray-activated cascaded generation of toxic $\bullet\text{OH}$ and CO molecules for oxygen-independent synergistic radiodynamic/gas therapy, as validated by a great number of *in vitro* and *in vivo* biological experiments. The zebrafish is not a new animal model as it has been widely used and reported in many papers. Herein, the optically transparent zebrafish larva was used to demonstrate the *in vivo* CO release from X-ray-activated HMOP-TBHP/ $\text{Fe}(\text{CO})_5$ and facilitate the observation of the generated CO signal under a confocal fluorescence microscope.

The reviewer's suggestion is very constructive, and we have simplified and clarified the main message in the revised manuscript to allow for the readers' quick and exact understanding.

3) It seems straightforward to use GC/MS or LC/MS to verify the mechanism of action of the novel photosensitizer.

Response: Thanks for the reviewer's kind suggestion. As explained above, only TBHP is a radiosensitizer while $\text{Fe}(\text{CO})_5$ is a typical CO-releasing molecule. As high-energy X-ray radiation can break the low-energy diselenide bond (Se-Se, 172 kJ/mol) to promote drug release (Ref. 34), it is reasonably speculated that the unstable peroxy bond (O-O, 146 kJ/mol) within TBHP would also be cleavable in the presence of X-ray radiation to generate hydroxyl radical ($\bullet\text{OH}$). In order to verify this mechanism of action of TBHP, the methylene blue (MB) dye was used to specifically probe the $\bullet\text{OH}$ generation from X-ray-activated TBHP because it could be bleached after selectively trapping $\bullet\text{OH}$ (Ref. 48). As shown in Fig. S16, the combination of TBHP and X-ray radiation caused a much faster decay of MB absorption than X-ray radiation alone, which indicates that the addition of TBHP accelerated the $\bullet\text{OH}$ production under the condition of X-ray radiation. The increased $\bullet\text{OH}$ yield only arose from the X-ray-activated O-OH bond cleavage within TBHP. Furthermore, terephthalic acid (TA) was employed to quantitatively measure the $\bullet\text{OH}$ yield based on the following mechanism. As 1.0 mol TA can chemically bind with 1.0 mol $\bullet\text{OH}$ radical to produce 1.0 mol 2-hydroxyterephthalic acid (TAOH), the generated concentration of $\bullet\text{OH}$ is equal to that of TAOH, which can be quantified by measuring its fluorescence emission intensity around 430 nm based on the standard curve (Ref. 49, 50). By calculation of the results in Figs. 4f-j, about 5 nmol/mL $\bullet\text{OH}$ was generated from the 72.2 $\mu\text{mol/mL}$ TBHP. According to the reviewer's kind suggestion, we tried to use the GC/MS to probe the change of TBHP after X-ray irradiation. However, as the $\bullet\text{OH}$ yield was very low (less than 1%), the GC/MS spectra of TBHP before and after X-ray irradiation failed to reflect such subtle change. Therefore, we can only use the very sensitive fluorescent probes to trap the small amount of $\bullet\text{OH}$ production. Herein, MB and TA were used to qualitatively and quantitatively probe the increased $\bullet\text{OH}$ yield, which unambiguously confirms the $\bullet\text{OH}$ generation from X-ray-activated TBHP and verifies this mechanism of action of TBHP.

Related references in the revised manuscript:

34. Ma, N., *et al.* Radiation-Sensitive Diselenide Block Co-Polymer Micellar Aggregates: Toward the Combination of Radiotherapy and Chemotherapy. *Langmuir* **27**, 5874-5878 (2011).
48. Satoh, A.Y., Trosko, J.E. & Masten, S.J. Methylene Blue Dye Test for Rapid Qualitative Detection of Hydroxyl Radicals Formed in a Fenton's Reaction Aqueous Solution. *Environ. Sci. Technol.* **41**, 2881-2887 (2007).
49. Chang, K., *et al.* Enhanced Phototherapy by Nanoparticle-Enzyme via Generation and Photolysis of Hydrogen Peroxide. *Nano Lett.* **17**, 4323-4329 (2017).
50. Son, H.Y., *et al.* Tannin-Titanium Oxide Multilayer as a Photochemically Suppressed Ultraviolet Filter. *ACS Appl. Mater. Inter.* **10**, 27344-27354 (2018).

4) Some of the language is a cliché. For example, consecutive sentences start with the following: 1) "In addition", 2) "Drawing inspiration", 3) "Of special note", 4) "Exhilaratingly", 5) "Extensively".

Response: Thanks for the reviewer's kind suggestion. All these cliché words have been deleted in the revised manuscript.

Reviewer #3 (Remarks to the Author):

In this study, Fan *et al.* employed an "ammonia-assisted hot water etching" strategy for the synthesis of small-sized (sub-50 nm) hollow mesoporous organosilica nanoparticles, and examined their co-delivery performance of tert-butyl hydroperoxide (TBHP) and iron pentacarbonyl (Fe(CO)₅) for oxygen-independent X-ray-activated synergistic therapy. Overall, the experiments were carefully conducted, the data was well analyzed, and the manuscript was well organized. However, there are still some significant problems in the current study that make it not suitable to be published in "Nature Communications".

1> For the "ammonia-assisted hot water etching" strategy highlighted by the authors in this study, actually, this strategy has been developed at least one decade ago for hollow materials synthesis (such as *Adv. Mater.* 2000, 12, 103-106; *Adv. Mater.* 2008, 20, 3987-4019; *ACS Nano*, 2010, 4, 529-539; *Adv. Healthcare Mater.* 2015, 4, 2797-2801). This is not a new method, the authors did not mention this in the manuscript. And even in the authors' recent paper (*ACS Nano*, 2018, 12, 1580-1591), they also used a similar method to fabricate sub-50 nm hollow organosilica nanoparticles.

Response: Thanks for the reviewer's comment. We respectfully disagree with reviewer that our proposed "ammonia-assisted hot water etching" strategy in this study has been developed one decade ago for synthesis of hollow silica materials. We think the reviewer might have misunderstood the novelty, advantage, and importance of our proposed "ammonia-assisted hot water etching" strategy and its unique featured application in the generic synthesis of a library of small-sized (sub-50 nm) hollow mesoporous organosilica nanoparticles (HMONs) with multiple framework hybridization of diverse organic moieties.

First, the literatures (such as *Adv. Mater.* 2000, 12, 103-106; *Adv. Mater.* 2008, 20, 3987-4019; *ACS Nano* 2010, 4, 529-539; *Adv. Healthcare Mater.* 2015, 4, 2797-2801) listed by the reviewer reported the synthesis of only large-sized (more than 200 nm) hollow mesoporous inorganic silica nanoparticles, but not small-sized (sub-50 nm) hollow mesoporous organosilica nanoparticles. Besides, the etching method involved in the above-mentioned four papers is called "ammonia-driven hydrothermal treatment", which is quite different from our proposed "ammonia-assisted hot water etching" strategy in this study. The complicated process of

“ammonia-driven hydrothermal treatment” must proceed in Teflon-lined autoclave at temperature higher than the boiling point (100 °C) of water, while our proposed convenient “ammonia-assisted hot water etching” strategy can work at 95°C without needing Teflon-lined autoclave.

Second, in our previous paper (*ACS Nano* 2018, 12, 1580-1591), only one type of sub-50 nm HMON with framework hybridization of only one thioether moiety was synthesized to encapsulate O₂-saturated perfluoropentane (PFP) for radiosensitization. But in this study, we first claim that this “ammonia-assisted hot water etching” strategy has the potential to act as a versatile method for the generic synthesis of a library of small-sized (sub-50 nm) HMONs with multiple framework hybridization of diverse organic moieties by changing only the introduced bissilylated organosilica precursors. Besides, the key reaction parameters (*e.g.*, catalyst amount, reaction temperature, etching direction/time, *etc.*) to this unique “ammonia-assisted hot water etching” strategy were clarified in detail, which may shed significant guidance on the generic synthesis of sub-50 nm multiple-hybridized HMONs with high reproducibility. Herein, in order to validate the general applicability of this strategy, other two bissilylated organosilica precursors, *i.e.*, bis-(triethoxysilyl)ethane (BTEE), bis-(triethoxysilyl)ethylene (BTEEE) have been used in combination with the original bis[3-(triethoxysilyl)propyl]tetrasulfide (BTES) and bis(triethoxysilyl)benzene (BTEB) to synthesize mono, double, triple and even quadruple-hybridized sub-50 nm HMONs with framework incorporation of diverse organic moieties (thioether, phenylene, ethane, ethylene). In total, ten types of sub-50 nm HMONs with uniform hollow-structured spherical morphology were successfully produced: four types of mono-hybridized HMON with framework incorporation of one kind of moiety (thioether, phenylene, ethane, or ethylene) through the use of BTES, BTEB, BTEE or BTEEE precursor; three types of double-hybridized HMON with framework incorporation of two kinds of moieties (thioether/phenylene, phenylene/ethane, or ethane/ethylene) through the use of mixed BTES/BTEB, BTEB/BTEE, or BTEE/BTEEE precursors; two types of triple-hybridized HMON with framework incorporation of three kinds of moieties (thioether/phenylene/ethane, or thioether/phenylene/ethylene) through the use of mixed BTES/BTEB/BTEE, or BTES/BTEB/BTEEE precursors; quadruple-hybridized HMON with framework incorporation of four kinds of moieties (thioether/phenylene/ethane/ethylene) through the use of mixed BTES/BTEB/BTEE/BTEEE precursors.

Third, the above sub-50 nm multiple-hybridized HMONs with framework incorporation of diverse organic moieties are expected to find wider applications in various fields, such as catalytic reaction, separation, and energy storage, not just limited to biomedicine. As an example, this study explored the new application of sub-50 nm biodegradable thioether-hybridized HMON in co-delivering *tert*-butyl hydroperoxide (TBHP) and iron pentacarbonyl (Fe(CO)₅) for X-ray-activated cascaded generation of toxic •OH and CO molecules for oxygen-independent synergistic radiodynamic/gas therapy, which has not been reported by other literatures so far.

2> For the synthesis of smaller-sized hollow mesoporous organosilica nanoparticles, there have already various one-step methods for hollow mesoporous organosilica nanoparticles synthesis, such as *Chem. Commun.*, 2016, 52, 3544-3547; *J. Mater. Sci.* 2017, 52, 2868-2878. Therefore, combined the 1st and 2nd comments, I did not find any new points for the materials synthesis (especially small-sized hollow mesoporous organosilica) in this study.

Response: Thanks for the reviewer’s comment. We cannot fully agree with reviewer that there have already been various one-step methods for the synthesis of small-sized hollow mesoporous

organosilica nanoparticles. Although the literatures (*Chem. Commun.* 2016, 52, 3544-3547; *J. Mater. Sci.* 2017, 52, 2868-2878) reported a one-step method for synthesis of HMONs, the sizes of these HMONs are much larger than 50 nm. The small-sized (sub-50 nm) HMONs can achieve the win-win between the decreased RES (reticuloendothelial system) uptake and the increased EPR (enhanced permeability and retention) effect for considerable tumor accumulation. Therefore, the controlled synthesis of sub-50 nm HMONs promises significant value to the development of high-performance silica-based nanocarriers for drug delivery. The sub-50 nm HMONs cannot be obtained by using the one-step method involved in the above-mentioned two papers, but can be well prepared by using our proposed unique “ammonia-assisted hot water etching” strategy in this study. Besides, only one special kind of bissilylated organosilica precursor, *i.e.*, 1,2-Bis(triethoxysilyl)ethane, is applicable to this limited one-step method (reported in the above-mentioned two papers) for the preparation of HMON with framework hybridization of only one ethane moiety. By comparison, our study proposes the well-established “ammonia-assisted hot water etching” strategy for the generic synthesis of a library of sub-50 nm HMONs with multiple organosilica framework hybridization. Almost all types of bissilylated organosilica precursors are applicable to this unique strategy, which is generally suitable for the preparation of sub-50 nm multiple-hybridized HMONs with framework incorporation of diverse organic moieties by changing only the bissilylated organosilica precursors. In total, ten types of sub-50 nm HMONs with uniform hollow-structured spherical morphology were successfully produced: four types of mono-hybridized HMON with framework incorporation of one kind of moiety (thioether, phenylene, ethane, or ethylene) through the use of BTES, BTEB, BTEE or BTEEE precursor; three types of double-hybridized HMON with framework incorporation of two kinds of moieties (thioether/phenylene, phenylene/ethane, or ethane/ethylene) through the use of mixed BTES/BTEB, BTEB/BTEE, or BTEE/BTEEE precursors; two types of triple-hybridized HMON with framework incorporation of three kinds of moieties (thioether/phenylene/ethane or thioether/phenylene/ethylene) through the use of mixed BTES/BTEB/BTEE or BTES/BTEB/BTEEE precursors; quadruple-hybridized HMON with framework incorporation of four kinds of moieties (thioether/phenylene/ethane/ethylene) through the use of mixed BTES/BTEB/BTEE/BTEEE precursors.

To the best of our knowledge, our proposed “ammonia-assisted hot water etching” strategy in this study is a unique versatile method for the generic synthesis of a library of sub-50 nm HMONs with uniform spherical morphology, high dispersity/stability and multiple framework hybridization of diverse organic moieties, which will substantially broaden the practical applications of sub-50 nm HMONs by incorporating the required organic moiety into the organosilica framework using the proper bissilylated organosilica precursor.

3> As for the hollow nanoparticles used in this study, the authors mentioned that “hollow cavity to allow for a large loading capacity of drugs”, but actually in this study, the loading capacity of Fe(CO)₅ is only 3.2 wt.%. I am not sure if it is necessary to use hollow nanoparticles in this study. If possible, please provide a comparison of drug loading between hollow and non-hollow nanoparticles.

Response: Thanks for the reviewer’s kind suggestion. Many literatures (Ref. 53, 54) have shown that hollow silica nanoparticles can load much more drugs (especially hydrophobic drugs) than non-hollow silica nanoparticles. According to the reviewer’s kind suggestion, we have measured TG curves of non-hollow organosilica nanoparticles (MSN@MONs without ammonia etching) before and after loading Fe(CO)₅, and the result (Fig. S29) shows that the loading capacity of

Fe(CO)₅ in MSN@MONs is about 1.5 wt.%, which is smaller than 3.2 wt.% of HMOP. The larger Fe(CO)₅ loading capacity of HMOP than MSN@MONs should be attributed to reason that the hollow cavity provides additional room for loading more hydrophobic molecules through hydrophobic-hydrophobic interactions. Besides, the hollow cavity of HMOP also offers enough space for efficient encapsulation of TBHP through hydrogen bonding force. Therefore, we used both the mesoporous channels and hollow cavity of HMOP to co-deliver Fe(CO)₅ and TBHP.

Related references in the revised manuscript:

53. Tang, F., Li, L. & Chen, D. Mesoporous Silica Nanoparticles: Synthesis, Biocompatibility and Drug Delivery. *Adv. Mater.* **24**, 1504-1534 (2012).

54. Chen, Y., *et al.* Engineering Inorganic Nanoemulsions/Nanoliposomes by Fluoride-Silica Chemistry for Efficient Delivery/Co-Delivery of Hydrophobic Agents. *Adv. Funct. Mater.* **22**, 1586-1597 (2012).

4> What is the loading capacity of TBHP?

Response: Thanks for the reviewer's question. The optimal TBHP loading capacities of both HMOP and HMOP-Fe(CO)₅ are 0.65 wt.% with the aim of avoiding the cytotoxicity arising from higher concentration of TBHP. Figs. S20, S21, S32, and S33 showed that the 0.65 wt.% loading capacity made HMOP-TBHP and HMOP-TBHP/Fe(CO)₅ exhibit high biocompatibility with negligible cytotoxicity.

5> Please provide more details of degradation study, such as temperature, particle concentration, TEM sample collection method, etc. And what is the degradation rates as a function of time?

Response: Thanks for the reviewer's kind suggestion. The detailed procedures for the biodegradation study of HMON have been added to the "Methods" section in the revised manuscript. As read in the "Methods" section: "**Biodegradation evaluation of HMON (with thioether hybridization):** To mimic the reductive tumor microenvironment, the biodegradation behavior of HMON was evaluated in both PBS and simulated body fluid (SBF) containing 10 mM GSH. 6 mg of HMON was dispersed in both 30 mL of PBS with 10 mM GSH and 30 mL of SBF with 10 mM GSH for incubation at 37 °C under slow stirring (300 rpm). Both the concentrations of HMON in PBS and SBF were 0.2 mg/mL. After different durations (1, 3, 5, 7, 10, and 14 d) of incubation, 1 mL of PBS and 1 mL of SBF were taken out and centrifuged to collect the partially biodegraded HMON. Subsequently, both the TEM characterization and ICP-AES analysis were performed to observe the degradation behavior of HMON and evaluate the degradation rate, respectively." As shown by the degradation curves in Figs. 4e and S15e, over 70% HMON was degraded after 14 days of incubation in PBS with 10 mM GSH and SBF with 10 mM GSH.

6> Since the particles were administrated by intravenous injection, please also provide the degradation data of HMONs in simulated body fluid or blood fluid.

Response: Thanks for the reviewer's kind suggestion. According to the above procedures, the biodegradation behavior of HMON in SBF (with 10 mM GSH) was evaluated by both TEM characterization and ICP-AES analysis. As seen from the degradation curves of HMON in Fig. S15e, over 70% HMON was degraded after 14 days of incubation in SBF with 10 mM GSH.

7> In this study, the authors mentioned the loading details of TBHP and Fe(CO)₅ into HMONs separately, but actually, there is almost no loading information for the resultant HMOP-TBHP/Fe(CO)₅ material, what is the loading amount of TBHP and Fe(CO)₅ in HMOP-TBHP/Fe(CO)₅? What is the detailed loading protocol? And how about the stability of HMOP-

TBHP/Fe(CO)₅ since the loading is mainly relying on noncovalent interactions (hydrogen bonding force and hydrophobic-hydrophobic interactions)?

Response: Thanks for the reviewer's questions. The loading percentages of TBHP and Fe(CO)₅ in HMOP were about 3.2 wt.% and 0.65 wt.%, respectively. The HMOP-TBHP/Fe(CO)₅ was prepared by sequential loading of Fe(CO)₅ and TBHP into HMOP. First, Fe(CO)₅ was adsorbed into the mesoporous channel and cavity of HMOP *via* hydrophobic-hydrophobic interaction. 20 mg of HMOP and 150 μL of Fe(CO)₅ stock solution were added to 15 mL of ethanol for 24 h of stirring. Then the HMOP-Fe(CO)₅ products were collected by centrifugation, washed with ethanol three times, and dispersed in 20 mL ultrapure water. Second TBHP was encapsulated into the cavity of HMOP through a fast vacuum impregnation method. 1 mg of HMOP-Fe(CO)₅ were put into a centrifuge tube (containing 1 mL of ultrapure water), followed by vacuum treatment. 10 μL of TBHP (diluted 1000 fold in ultrapure water) was quickly injected into the tube. Then the tube mouth was sealed tightly and subjected to ultrasonic treatment in the ice bath for 30s to allow the complete encapsulation of TBHP into the cavity of HMOP-Fe(CO)₅, which yielded the HMOP-TBHP/Fe(CO)₅ product. The above loading protocol has been added to the "Part A: Experimental procedures" section of the revised supporting information. According to the literatures (Ref. 35, 42, 52), hydrophobic drug molecules can be loaded into the mesopores and cavity of hollow-structured silica nanoparticles *via* hydrophobic-hydrophobic interaction, and little was released in PBS. In our study, the hydrophobic Fe(CO)₅ molecules could also be firmly loaded into HMOP *via* hydrophobic-hydrophobic interaction, which demonstrated high stability with little leakage (Figs. S27, S28). Besides, hydrophilic TBHP could be loaded into HMOP through hydrogen bonding force (between Si-OH and O-OH), and the stability of hydrogen bonding force with little leakage has been also validated by many literatures like Ref. 47. Therefore, the well-designed stable TBHP/Fe(CO)₅ co-loaded HMOP (i.e. HMOP-TBHP/Fe(CO)₅) could be used for the following X-ray-activated synergistic RDT/gas therapy.

Related references in the revised manuscript:

35. Jin, Z., *et al.* Intratumoral H₂O₂-triggered release of CO from a metal carbonyl-based nanomedicine for efficient CO therapy. *Chem. Commun.* **53**, 5557-5560 (2017).

42. Chen, Y., *et al.* Colloidal HPMO Nanoparticles: Silica-Etching Chemistry Tailoring, Topological Transformation, and Nano-Biomedical Applications. *Adv. Mater.* **25**, 3100-3105 (2013).

47. Zhang, K., *et al.* CO₂ bubbling-based 'Nanobomb' System for Targetedly Suppressing Panc-1 Pancreatic Tumor via Low Intensity Ultrasound-activated Inertial Cavitation. *Theranostics* **5**, 1291-1302 (2015).

52. Chen, Y., Chen, H. & Shi, J. *In Vivo* Bio-Safety Evaluations and Diagnostic/Therapeutic Applications of Chemically Designed Mesoporous Silica Nanoparticles. *Adv. Mater.* **25**, 3144-3176 (2013).

8> Hydroxyl radical (•OH) released from TBHP could further trigger Fe(CO)₅ to release CO molecules, this finding is very interesting in this study. However, as mentioned above, since there is multi-step protocol for producing HMOP-TBHP/Fe(CO)₅, how to keep the amount and ratio of TBHP and Fe(CO)₅ in a controllable, stable, and reproducible way?

Response: Thanks for the reviewer's question. HMOP-TBHP/Fe(CO)₅ was prepared by loading Fe(CO)₅ into the mesopores/cavity of HMOP through hydrophobic-hydrophobic interaction and then encapsulating TBHP into the cavity of HMOP through hydrogen bonding force. The preparation process contains only two steps for the co-delivery of TBHP and Fe(CO)₅ by HMOP. If the amount of each component (HMOP, TBHP, Fe(CO)₅), the volume of reaction solvent, and other experimental parameters are precisely controlled, the loading amount and ratio of TBHP and Fe(CO)₅ within HMOP can be kept the same for each preparation of HMOP-TBHP/Fe(CO)₅. First, for the reproducible loading of Fe(CO)₅ into HMOP, 20 mg of HMOP and 150 μL of

Fe(CO)₅ stock solution were added to 15 mL of ethanol for 24 h of stirring. Then the HMOP-Fe(CO)₅ products were collected by centrifugation, washed with ethanol three times, and dispersed in 20 mL ultrapure water. The as-synthesized HMOP-Fe(CO)₅ showed high stability (Figs. S27, S28) with the fixed 3.2 wt.% loading percentage of Fe(CO)₅ (Fig. 6f). Second, for the reproducible encapsulation of TBHP into stable HMOP-Fe(CO)₅, 1 mL of the above ultrapure water of HMOP-Fe(CO)₅ (1 mg/mL) was put into a centrifuge tube and subjected to vacuum treatment. 10 μL of TBHP (diluted 1000 fold by ultrapure water) was quickly injected into the tube. Then the tube mouth was sealed tightly and subjected to ultrasonic treatment in the ice bath for 30s to allow the complete encapsulation of TBHP (about 0.65 wt.% loading percentage) into the cavity of HMOP-Fe(CO)₅, which yielded the HMOP-TBHP/Fe(CO)₅ product for future use. Therefore, by strictly following the above two-step loading procedures, the HMOP-TBHP/Fe(CO)₅ can be easily reproduced with a stable co-loading ratio (0.65 wt.%/3.2 wt.%, 1/5) of TBHP/Fe(CO)₅ within HMOP.

9> In figure 7c, the HMOP-TBHP group should be provided.

Response: Thanks for the reviewer's kind suggestion. The confocal fluorescence images of living zebrafish larvae with brain microinjection of HMOP-TBHP with or without 6 Gy of X-ray irradiation have been added to Fig. 8c. Similar to the HMOP-Fe(CO)₅ group, the fluorescence signal of CO is hardly seen in the HMOP-TBHP group even upon X-ray irradiation, which indicates negligible CO release from HMOP-TBHP.

10> In Figures 3c and S21a, why the primary hollow nanoparticles turned out to be solid ones? Please clarify.

Response: Thanks for the reviewer's question. The break-up of disulfide bonds in the presence of reductive GSH was known to cause the degradation of thioether-hybridized organosilica nanoparticles (Ref. 44). During the degradation process, the dissolved products (*e.g.*, silicic acid, polysilicic acid, *etc.*) might enter and fill the cavity of some non-degraded HMOPs or regenerate some silica nanoparticles through the condensation of Si-OH groups (Ref. 45, 46), so some hollow nanoparticles seemed to turn into solid ones. However, these "solid-like" nanoparticles were temporary and could be finally degraded with the increasing time of incubation in GSH solution (Figs. 4a-e, S15). This clarification has been added to the revised manuscript.

Related references in the revised manuscript:

44. Zhou, M., *et al.* One-pot synthesis of redox-triggered biodegradable hybrid nanocapsules with a disulfide-bridged silsesquioxane framework for promising drug delivery. *J. Mater. Chem. B* **5**, 4455-4469 (2017).

45. Du, X., *et al.* Disulfide-Bridged Organosilica Frameworks: Designed, Synthesis, Redox-Triggered Biodegradation, and Nanobiomedical Applications. *Adv. Funct. Mater.* **28**, 1707325 (2018).

46. Chen, Y., *et al.* Reversible Pore-Structure Evolution in Hollow Silica Nanocapsules: Large Pores for siRNA Delivery and Nanoparticle Collecting. *Small* **7**, 2935-2944 (2011).

11> In Figure 8c-d, for better comparison, please also provide the results of other treatment groups.

Response: Thanks for the reviewer's kind suggestion. The results of p53 and caspase 3 expression in U87MG tumors subjected to other treatments have been added to Figs. 9c, d of the revised manuscript. It can be seen that the upregulation of p53 and caspase 3 expression demonstrates a positive correlation with the therapeutic effect of the treatment.

Therefore, based on the above considerations, there is limited new information in the materials synthesis part, the cascaded generation $\bullet\text{OH}$ and CO for multimodal synergistic therapy is a new finding, but the construction of HMOP-TBHP/ $\text{Fe}(\text{CO})_5$ particulate system is not clearly presented.

Response: Thanks for the reviewer's comment. As explained above, our proposed "ammonia-assisted hot water etching" strategy in this study may be developed as a unique versatile method for the generic synthesis of a library of sub-50 nm HMONS with multiple framework hybridization of diverse organic moieties, which has not been systemically reported by other papers. We appreciate the reviewer's viewpoint that the cascaded generation $\bullet\text{OH}$ and CO for multimodal synergistic therapy is a new finding. Overall, the specific originality and novelty of this work are clarified as follows.

(1) Generic synthetic methodology. A versatile "ammonia-assisted hot water etching" method is proposed for the generic synthesis of a library of small-sized (sub-50 nm) HMONS with multiple framework hybridization of diverse organic moieties by changing only the introduced bisilylated organosilica precursors (accepted by the reviewer #4, who said that "The authors claim reporting a generic method for synthesis of small-sized HMONS, demonstrated by fabrication of three different HMONS"). In total, ten types of sub-50 nm HMONS with uniform hollow-structured spherical morphology were successfully produced: four types of mono-hybridized HMION with framework incorporation of one kind of moiety (thioether, phenylene, ethane, or ethylene) through the use of BTES, BTEB, BTEE or BTEEE precursor; three types of double-hybridized HMION with framework incorporation of two kinds of moieties (thioether/phenylene, phenylene/ethane, or ethane/ethylene) through the use of mixed BTES/BTEB, BTEB/BTEE, or BTEE/BTEEE precursors; two types of triple-hybridized HMION with framework incorporation of three kinds of moieties (thioether/phenylene/ethane or thioether/phenylene/ethylene) through the use of mixed BTES/BTEB/BTEE or BTES/BTEB/BTEEE precursors; quadruple-hybridized HMION with framework incorporation of four kinds of moieties (thioether/phenylene/ethane/ethylene) through the use of mixed BTES/BTEB/BTEE/BTEEE precursors.

(2) First demonstration of oxygen-independent radiodynamic therapy (RDT). By making full use of the unique advantage of high-energy X-ray irradiation in cleaving the low-energy unstable peroxy bond, TBHP can be activated by X-ray to generate highly toxic $\bullet\text{OH}$ without reliance on oxygen, which gives rise to a novel treatment paradigm of RDT for effectively killing both normoxic and hypoxic cancer cells (accepted by the reviewer #1, who said that "The paper 'Generic synthesis of small-sized hollow mesoporous organosilica nanoparticles for oxygen-independent X-ray-activated synergistic therapy', by Chen et al, deals with the development of a new treatment paradigm of radiodynamic therapy").

(3) First realization of X-ray-activated synergistic therapy. On the basis of the cascaded generation of $\bullet\text{OH}$ and CO from TBHP/ $\text{Fe}(\text{CO})_5$ co-loaded PEGylated HMION through X-ray-activated sequential cleavage of O-OH bond and Fe-CO bond without oxygen dependence, this study provides a typical paradigm shifting nanotechnology in realizing the renaissance of oxygen-dependent RT into oxygen-independent synergistic RDT/gas therapy (accepted by the reviewer #3, who said that "the cascaded generation $\bullet\text{OH}$ and CO for multimodal synergistic therapy is a new finding"), thus overcoming the Achilles' heel of conventional RT.

Moreover, in order to clearly present the construction of HMOP-TBHP/ $\text{Fe}(\text{CO})_5$ particulate system, more experiments, characterizations, and discussions have been supplemented according to all the reviewers' kind suggestions. Therefore, we strongly believe that the well-established

generic synthetic method, the well-designed small-sized HMNs with multiple framework hybridization, together with the well-developed X-ray-activated oxygen-independent synergistic RDT/gas therapy in this novel multidisciplinary study will arouse the broad interest of the readership of *Nature Communications* from different fields, including material science, chemistry, radiology, nano-biotechnology, pharmacy and biomedicine.

Reviewer #4 (Remarks to the Author):

The authors report synthesis and biological evaluation of small sized hollow mesoporous organosilica nanoparticles (HMNs) encapsulating a radiosensitizer tert-butyl hydroperoxide (TBHP) and iron pentacarbonyl ($\text{Fe}(\text{CO})_5$) for a novel type cancer therapy, which combines radiodynamic therapy (X-ray activated hydroxy radical generation) and a CO gas therapy. Several hydroxy radical-forming radiosensitizers are currently in clinical trials for enhancing efficacy of radiotherapy. In this paper X-ray activated hydroxy radical generation and CO gas therapy are delicately combined into a same nanoformulation in a novel way, leading to excellent tumor growth inhibition when evaluated in U87MG tumor-bearing mice. These results may lead to more efficient anti-cancer therapies and encourage discovery of other dynamic combination treatments. In order to be able to evaluate applicability and significance of the presented results the following comments should be addressed:

-The authors claim reporting a generic method for synthesis of small-sized HMNs, demonstrated by fabrication of three different HMNs. More comprehensive characterization of the fabricated materials should be presented for facilitating comparison of the different hybridized HMNs; such as mesopore size for all particle types and elemental analysis for quantification of the organic moieties. The authors are asked to report also the BTES/BTEB ratio on the dual hybridized HMNs.

Response: Thanks for the reviewer's kind suggestion. In order to validate the unique advantage of our proposed "ammonia-assisted hot water etching" method in the generic synthesis of a library of small-sized (sub-50 nm) HMNs with multiple framework hybridization of diverse organic moieties, other two bisilylated organosilica precursors, *i.e.*, bis-(triethoxysilyl)ethane (BTEE), bis-(triethoxysilyl)ethylene (BTEEE) were used in combination with BTES and BTEB to synthesize more types of mono, double, triple and even quadruple-hybridized sub-50 nm HMNs with framework incorporation of more kinds of organic moieties (thioether, phenylene, ethane, ethylene). In total, ten types of sub-50 nm HMNs with uniform hollow-structured spherical morphology have been successfully produced: four types of mono-hybridized HMN with framework incorporation of one kind of moiety (thioether, phenylene, ethane, or ethylene), three types of double-hybridized HMN with framework incorporation of two kinds of moieties (thioether/phenylene, phenylene/ethane, or ethane/ethylene), two types of triple-hybridized HMN with framework incorporation of three kinds of moieties (thioether/phenylene/ethane or thioether/phenylene/ethylene), and quadruple-hybridized HMN with framework incorporation of four kinds of moieties (thioether/phenylene/ethane/ethylene). The TEM images, particle size distributions, Raman spectra, BET surface areas, mesopore size distributions, and EDS spectra (elemental analyses) have been done for all the as-synthesized ten types of sub-50 nm HMNs, which are presented in Figs 2-3, S5-14. Besides, the ratios of the introduced bisilylated

organosilica precursors for the production of the ten types of sub-50 nm HMONS have been also added to the “Methods” section of the revised manuscript.

-Fe(CO)₅ was adsorbed into the mesoporous channel via weak interactions. In Fig S19 the authors demonstrate stability of the Fe(CO)₅ adsorption by inspecting color of the supernatant. More persuasive evaluation of Fe(CO)₅ leakage should be presented. Please, quantify the possible Fe(CO)₅ release at different time points in the storage buffer and also in serum.

Response: Thanks for the reviewer’s kind suggestion. Many literatures (Ref. 35, 42, 52) have reported that hydrophobic drug molecules could be loaded into the mesopores and cavity of hollow-structured silica nanoparticles *via* hydrophobic-hydrophobic interaction, and little was released in PBS. We have also quantified the released Fe(CO)₅ (during 24 h of incubation in PBS and serum) by measuring the Fe concentration *via* ICP-OES. The Fe(CO)₅ releasing profiles (Fig. S28) show that less than 2% Fe(CO)₅ was released during 24 h in PBS and serum, which indicates the high stability of HMOP-Fe(CO)₅ with little leakage.

Related references in the revised manuscript:

35. Jin, Z., *et al.* Intratumoral H₂O₂-triggered release of CO from a metal carbonyl-based nanomedicine for efficient CO therapy. *Chem. Commun.* **53**, 5557-5560 (2017).

42. Chen, Y., *et al.* Colloidal HPMO Nanoparticles: Silica-Etching Chemistry Tailoring, Topological Transformation, and Nano-Biomedical Applications. *Adv. Mater.* **25**, 3100-3105 (2013).

52. Chen, Y., Chen, H. & Shi, J. *In Vivo* Bio-Safety Evaluations and Diagnostic/Therapeutic Applications of Chemically Designed Mesoporous Silica Nanoparticles. *Adv. Mater.* **25**, 3144-3176 (2013).

-Figure 3i: The authors should explain why the fluorescence intensity is lower for HMOP-TBHPs even without X-ray irradiation?

Response: Thanks for the reviewer’s question. As reported in the literatures (Ref. 49, 50), terephthalic acid (TA) is usually used for the detection of •OH radicals as nonfluorescent TA can be oxidized by •OH to generate fluorescent 2-hydroxyterephthalic acid (TAOH). Therefore, the fluorescence intensity of TAOH (oxidized TA) is an indicator of the concentration of generated •OH radicals. As HMOP-TBHP itself did not generate •OH and only yielded •OH when X-ray activated the cleavage of the O-OH bond within TBHP, the fluorescence intensity of TAOH was much lower for HMOP-TBHP without X-ray irradiation than that with X-ray irradiation. This explanation has been added to the revised manuscript.

Related references in the revised manuscript:

49. Chang, K., *et al.* Enhanced Phototherapy by Nanoparticle-Enzyme via Generation and Photolysis of Hydrogen Peroxide. *Nano Lett.* **17**, 4323-4329 (2017).

50. Son, H.Y., *et al.* Tannin–Titanium Oxide Multilayer as a Photochemically Suppressed Ultraviolet Filter. *ACS Appl. Mater. Inter.* **10**, 27344-27354 (2018).

-The authors claimed that biosafety of HMOP-TBHP/Fe(CO)₅ was confirmed by evaluating the H&E stained sections of major organs after iv-injection of the particles. The *in vivo* biodistribution study of ⁶⁴Cu-labeled HMOPs demonstrate that there is major accumulation of the particles in liver and spleen. Cell viability was tested only in U87MG cells (Fig. S22). Cell viability in hepatic cells (e.g. HepG2) and in RAW macrophages should be presented.

Response: Thanks for the reviewer’s kind suggestion. The cytotoxicities of HMOP-TBHP, HMOP-Fe(CO)₅, and HMOP-TBHP/Fe(CO)₅ against HepG2 cells and RAW macrophage cells have been evaluated. Figs. S21 and S33 show that these two cell lines still exhibit much high viability after incubation with HMOP-TBHP, HMOP-Fe(CO)₅, and HMOP-TBHP/Fe(CO)₅ for 24 h, which indicates the great biocompatibility of these nanomaterials.

-Fig 7b: add %ID/g in lung and blood at different time-points.

Response: Thanks for the reviewer's kind suggestion. The biodistribution of ^{64}Cu -labeled HMOP (%ID/g) in lung and blood at different time points has been added to Fig. 8b of the revised manuscript.

-Description of the Cu-64 radiolabeling procedure should be added to the supporting information, including also evaluation of stability of the radiolabel (i.e. possible Cu-64 release) at different time points in PBS (pH 7.4) and serum.

Response: Thanks for the reviewer's kind suggestion. The experimental procedures of the ^{64}Cu radiolabeling method have been added to the "Part A: Experimental procedures" section of the revised supporting information. As read in the "Part A: Experimental procedures" section: "**Preparation of ^{64}Cu -labeled HMOP.** First, the thiol group was modified onto the surface of HMOP. 20 mg of HMOP was dissolved in 30 mL of ethanol, followed by the addition of 150 μL of MPTES and 200 μL of NH_4OH . The mixed solution was stirred for 10 h, and the product thiol functionalized HMOP (HMOP-SH) was obtained by centrifugation and washing with ethanol several times. Second, ^{64}Cu was used to label HMOP-SH by taking advantage of the strong chelating affinity of thiol group towards radionuclides. 2 μL of $^{64}\text{CuCl}_2$ (3~5 mCi) was added to 0.5 mL of MES buffer (10 mM, pH 7.3) for 1-2 minutes of incubation. Thereafter, 0.5 mCi $^{64}\text{CuCl}_2$ in MES buffer was added to a vial containing MES buffer of HMOP-SH, and the reaction was heated to 70 $^\circ\text{C}$ for 45~60 minutes. After cooling down to the ambient temperature, an aliquot of ^{64}Cu -labeled HMOP was taken for determination of the radiochemical purity by radioTLC using 0.1 M Citric acid (pH 5) as a development solvent and iTLC plates (Fisher Scientific). Rf of ^{64}Cu -labeled HMOP is 0~0.1, and Rf of free ^{64}Cu is 0.9.

The experimental procedures of evaluation of the radiolabeling stability have been also added to the "Part A: Experimental procedures" section of the revised supporting information. As read in the "Part A: Experimental procedures" section: "**Evaluation of the radiolabeling stability of ^{64}Cu -labeled HMOP.** 20 μL of ^{64}Cu -labeled HMOP in MES buffer was added to 200 μL of PBS (pH 7.4) and serum, respectively. The PBS solution of ^{64}Cu -labeled HMOP was incubated at room temperature, whereas the serum of ^{64}Cu -labeled HMOP was incubated at 37 $^\circ\text{C}$. At 1, 4, and 24 h time points, 2 μL aliquots of ^{64}Cu -labeled HMOP were taken from PBS and serum, and then loaded on iTLC plates for measuring the radiochemical yield. Rf of ^{64}Cu -labeled HMOP is 0~0.1, and Rf of free ^{64}Cu is 0.9".

As shown in Fig. S40, the radiochemical yield of thiol functionalized HMOP (HMOP-SH) is almost 100%, which indicates the strong binding affinity of HMOP-SH towards ^{64}Cu and the high radiolabeling efficiency. Moreover, the ^{64}Cu -labeled HMOP demonstrates high radiolabeling stability in PBS for at least 24 h (Fig. S41a₁-a₃). Although only a little free ^{64}Cu was released when incubating ^{64}Cu -labeled HMOP in serum for 24 h (Fig. S41b₃), negligible free ^{64}Cu was disassociated from thiol functionalized HMOP in serum for at least 4 h (Fig. S41b₁, b₂), which also indicates the relatively high radiolabeling stability of ^{64}Cu -labeled HMOP in serum.

-The authors should report number of animals used for in vivo toxicity evaluation and for each treatment group in evaluation of the therapeutic effect in U87MG tumors. Number of the animals should be included also to all figure captions reporting any in vivo data.

Response: Thanks for the reviewer's kind suggestion. Five mice per group were used for the *in vivo* toxicity evaluation and therapy experiments. Besides, the numbers of mice have been added

to all figure captions containing any *in vivo* data in the revised manuscript and supporting information.

-Line 201 and Fig S8: Explain MB.

Response: Thanks for the reviewer's kind suggestion. MB is the abbreviation for the dye "methylene blue". MB is a selective probe for trapping $\bullet\text{OH}$ (Ref. 48). As $\bullet\text{OH}$ is able to cause the MB absorption decay, the generated $\bullet\text{OH}$ amount can be roughly estimated by the decay ratio of MB absorption. This related explanation has been added to Fig. S16 in the revised supporting information.

Related references in the revised manuscript:

48. Satoh, A.Y., Trosko, J.E. & Masten, S.J. Methylene Blue Dye Test for Rapid Qualitative Detection of Hydroxyl Radicals Formed in a Fenton's Reaction Aqueous Solution. *Environ. Sci. Technol.* **41**, 2881-2887 (2007).

-Details of the X-ray source and absorbed dose calculation for the cell studies and *in vivo* experiments should be provided.

Response: Thanks for the reviewer's kind suggestion. The details of the X-ray source and radiation doses for the *in vitro* and *in vivo* experiments have been added to the "Part A: Experimental procedures" section of the revised supporting information. The MultiRad 225, a cabinet x-ray system created by Faxitron Bioptics LLC, was used to provide X-ray radiation for the *in vitro* and *in vivo* experiments. The working voltage, tube current, and filter type were set as 224 kV, 17.8 mA, and 0.5 mm Aluminum, respectively. The irradiation area is designated by the circular outlines on the turntable. There are seven shelves which are corresponding to the seven circular outlines on the turntable. The turntable was put on the Shelf 5 with the maximum irradiation diameter of 20.4 cm. First, before the experiment, we selected the "ADC Program" to calculate the time needed for the accumulated dose (measured by the dosimeter) to reach the target dosage. Second, we selected the "Manual Mode" and placed the cells or mice within this maximum irradiation circle (20.4 cm in diameter). Third, the non-irradiated cells and other body parts (except tumor) of mice were blocked by the lead plates to make sure that only the irradiated cells and tumor of mice were exposed to X-ray radiation. Fourth, we input the irradiation time recorded in the first step and started the X-ray. After the X-ray was completed, the radiation dose received by the irradiated cells and tumor of mice (i.e. absorbed dose) was equal to the target dosage. For example, when the target dosage was set as 8 Gy during the *in vivo* experiment, the time recorded was 46.8s, and then the absorbed dose by the tumor of mice was 8 Gy.

REVIEWERS' COMMENTS:

Reviewer #1

As no technical concerns were raised in the referees previous review and publication was recommended. It was decided not to consult the reviewer for this round of review.

Reviewer #3

In comments to the editor reviewer #3 questioned the novelty claims around the synthesis, referencing the author's previous work, but commented the application was interesting and comprehensive and recommended publication based on this.

Reviewer #2 (Remarks to the Author):

The authors have provided a comprehensive response to all issues raised.

Reviewer #4 (Remarks to the Author):

All my comments were addressed and comprehensively answered. No any further comments.